# DHX36 binding induces RNA structurome remodeling and regulates RNA abundance via m⁶A reader YTHDF1

Yuwei Zhang[1,9], Jieyu Zhao[2,9], Xiaona Chen[3,4], Yulong Qiao[3,4], Jinjin Kang[5], Xiaofan Guo[3,4], Feng Yang[1], Kaixin Lyu [2], Yiliang Ding [6], Yu Zhao[5], Hao Sun [7] ✉, Chun-Kit Kwok [2,8] ✉ & Huating Wang [3,4] ✉

RNA structure constitutes a new layer of gene regulatory mechanisms. RNA binding proteins can modulate RNA secondary structures, thus participating in post-transcriptional regulation. The DEAH-box helicase 36 (DHX36) is known to bind and unwind RNA G-quadruplex (rG4) structure but the transcriptome-wide RNA structure remodeling induced by DHX36 binding and the impact on RNA fate remain poorly understood. Here, we investigate the RNA structurome alteration induced by DHX36 depletion. Our findings reveal that DHX36 binding induces structural remodeling not only at the localized binding sites but also on the entire mRNA transcript most pronounced in 3'UTR regions. DHX36 binding increases structural accessibility at 3'UTRs which is correlated with decreased post-transcriptional mRNA abundance. Further analyses and experiments uncover that DHX36 binding sites are enriched for N6-methyladenosine (m⁶A) modification and YTHDF1 binding; and DHX36 induced structural changes may facilitate YTHDF1 binding to m⁶A sites leading to RNA degradation. Altogether, our findings uncover the structural remodeling effect of DHX36 binding and its impact on RNA abundance through regulating m⁶A dependent YTHDF1 binding.

In recent years, interest in RNA secondary structure has exploded due to its implications in almost all biological functions and its newly appreciated capacity as a therapeutic agent/target[1]. Examples of secondary structures include long-range interactions, hairpins, R-loops, and G-quadruplexes (G4), and they are formed through interactions of non-adjacent nucleotides. Due to their large size, the structures of mRNAs are challenging to study, but the past decade has seen rapid development in RNA structure-probing methods to capture transcriptome-wide RNA structures (that is, RNA structurome) in many species and across conditions[1]. For example, structure-seq[2] was developed for mapping in vivo RNA structures and interactions by coupling chemical structure probing with deep sequencing. Chemical modification on the unpaired bases leads to the reverse transcription (RT) stops, which can be read out as a reactivity score and be used to infer RNA structures. It provides quantitative RNA secondary structural information across thousands of transcripts at nucleotide

[1]Department of Chemical Pathology, Li Ka Shing Institute of Health Sciences, Chinese University of Hong Kong, Hong Kong, SAR, China. [2]Department of Chemistry and State Key Laboratory of Marine Pollution, City University of Hong Kong, Hong Kong, SAR, China. [3]Department of Orthopaedics and Traumatology, Li Ka Shing Institute of Health Sciences, Chinese University of Hong Kong, Hong Kong, SAR, China. [4]Center for Neuromusculoskeletal Restorative Medicine Limited, Hong Kong, SAR, China. [5]Molecular Cancer Research Center, School of Medicine, Shenzhen Campus of Sun Yat-sen University, Sun Yat-sen University, Shenzhen, China. [6]Department of Cell and Developmental Biology, John Innes Centre, Norwich Research Park, Norwich NR4 7UH, United Kingdom. [7]Warshel Institute for Computational Biology, The Chinese University of Hong Kong, Shenzhen, China. [8]Shenzhen Research Institute of City University of Hong Kong, Shenzhen, China. [9]These authors contributed equally: Yuwei Zhang, Jieyu Zhao. ✉e-mail: sunhao100@cuhk.edu.cn; ckkwok42@cityu.edu.hk; huating.wang@cuhk.edu.hk

resolution. Several other probing methods have also been developed[3–5], thus providing tools for systematic interrogation of in vivo RNA structures.

The roles of RNA secondary structures in key biological functions can be seen in every type of RNA mainly because of their importance in mediating RNA association with RNA binding proteins (RBPs)[6–9]. During the mRNA lifecycle, RBPs regulate diverse transcriptional and post-transcriptional stages. They can bind to pre-mRNA molecules in the nucleus and regulate its maturation and transportation to the cytoplasm, where they regulate translation and degradation. Emerging evidence from transcriptome wide profiling of RBP binding such as crosslinking immunoprecipitation sequencing (CLIP-seq) and RNA bind-n-seq (RBNS) suggests that other than RNA sequence motifs, local RNA secondary structures play a determinant role in RBP binding and regulation[10–13]. Still, the field remains largely nascent, and many key questions need to be answered, such as to what degree and how RNA secondary structures contribute to RBP regulation of post-transcriptional RNA processing?

The interplay between RBPs and RNA structures is made more complex by the fact that RBP binding also orchestrates secondary structure formation and stabilization. Of note, RNA structure formation is, to a large extent, modulated by helicases that can bind and rearrange RNA structures and ribonucleoprotein complexes, thus playing crucial roles in transcriptional, post-transcriptional, and translational regulation[7,14]. For example, DHX36 is a DEAH-Box helicase which can bind and unwind RNA G-quadruplex (rG4s) and RNA duplex via a conserved DHX36-specific motif (DSM) and other auxiliary domains[15,16]. rG4 structures are non-canonical secondary structures formed in G-rich RNA sequences by the stacking of guanine-quartets through Hoogsteen hydrogen bonding[17]. Recent studies suggest that rG4 formation can modulate (either favor or block) in vitro RBP binding to mRNA molecules[18,19]. Therefore, by unwinding rG4s, DHX36 plays an essential role in post-transcriptional regulation at many levels[15,20–23]. For example, in our recent study[22], by CLIP-seq profiling, we found that DHX36 primarily binds with rG4s in myoblast cells, and unwinding 5′UTR rG4s endows its ability to promote mRNA translation. Nevertheless, this study also revealed its binding at CDSs, 3′UTRs, and introns, implying the potential functions of DHX36 in regulating mRNA abundance as well as other aspects of mRNA metabolism. Apart from rG4-dependent mechanisms, an in vitro study based on the crystal structure of DHX36 protein revealed that DHX36 can also bind to and unwind RNA duplexes[15]. Therefore, it is imperative to illuminate how DHX36 binding induces RNA structural changes system-wide and how the changes control post-transcriptional mRNA processing.

The RNA structure/RBP interplay becomes more intriguing by the recent discovery of the interplay between RNA structures, RNA modifications, and RBP binding[13]. N6-methyladenosine (m6A) modification is the most abundant internal modification in mRNAs and contributes to diverse fundamental cellular functions, which is largely dependent on the "reader" proteins that recognize and bind with m6A sites[24]. Among them, the YTHDF1 reader is known for regulating mRNA degradation and translation via its m6A binding ability[25–28]. Interestingly, m6A modification can alter the local structures in RNAs via the so-called "m6A-switch" to modulate the binding of RBPs[13,29]. Mutually, RNA structural accessibility also orchestrates the abundance and effects of m6A, because the binding activities of "writer" complex METTL3/METTL14[30], "erasers" FTO and ALKBH5[31,32], and some "readers"[13,29] are sensitive to target RNA structures. YTHDF readers appear to prefer unstructured RNAs[33], but it remains unclear whether the unwinding of RNA structures is essential for the binding of these proteins to m6A sites.

In this study, we investigated how DHX36 binding influences mRNA secondary structures in vivo and the mechanism underlying DHX36-induced structural changes in post-transcriptional regulation. By conducting the RNA structure-seq in wild-type (WT) and DHX36 knockout (DHX36-KO) HEK293T cells, we found that DHX36 depletion induces a global increase of RNA structures. Combining with DHX36 binding profiles, we found that DHX36 binding increases RNA structure accessibility, especially in 3′UTRs. Further analyses demonstrate that induced accessible structures in 3′UTRs are related to post-transcriptional mRNA decrease. Moreover, m6A-dependent YTHDF1 binding events are enriched in the DHX36 binding sites and may mediate DHX36 regulation of mRNA abundance. Additional experimental assays were conducted on selected target mRNAs to substantiate that DHX36 binding promotes YTHDF1 binding to m6A sites and subsequent mRNA degradation. Altogether, our findings uncover transcriptome-wide structural changes induced by DHX36 binding and illuminate how the changes orchestrate mRNA abundance.

## Results

### In vivo RNA structurome profiling unveils DHX36 depletion-induced global increase of RNA structures

To have a thorough understanding of DHX36 function in modulating RNA structures[15], we deleted DHX36 in HEK293T cells using CRISPR/Cas9-mediated gene editing to obtain a DHX36-KO cell line (Fig. 1A) and performed a slightly modified SHAPE-based Structure-seq[34] in WT and KO cells. 2-methylnicotinic acid imidazolide (NAI) but not dimethyl sulfide (DMS) was used therefore, all four RNA bases were modified (Fig. 1B). NAI treatment can acylate the 2′hydroxyl group of all four RNA bases not involved in base pairing, thus blocking the reverse transcription (RT) at these positions and generating discriminative signals of RT stops in sequencing data (Fig. 1C). Two replicates were performed for each sample and high reproducibility was observed when correlating the RT stops of each nucleotide between the two replicates ($r > 0.9$, Fig. S1A). After removing transcripts with low RT stop coverage, the analysis yielded 22,390 mRNAs with at least 1 RT stop per nucleotide (Supplementary Data 1). SHAPE reactivity of each nucleotide was then calculated by subtracting the number of RT stops of the mock (DMSO)-treated sample from the NAI-treated sample and normalized to yield an accurate assessment of RNA accessibility (see Methods)[35] (Fig. S1B). As additional validation to the high quality of our data, the SHAPE reactivity profiles of 18S rRNA in both WT (Fig. S1C, D) and KO (Fig. S1E, F) were in concordance with the phylogenetically derived structures, indicating that our data can accurately reflect in vivo RNA secondary structures.

Next, to examine the effect of DHX36 loss on RNA structures, an average value and a Gini index of SHAPE reactivity scores were calculated. A higher Gini index is correlated with a more heterogeneous distribution of SHAPE reactivity scores, thus increased the structurization of the entire mRNA[3,36]. As shown in Fig. 1D, E, the average SHAPE reactivity scores of mRNAs were significantly decreased (FC = 0.74, $P < 2.23e-308$) in the KO vs. WT cells; a concomitant increase of Gini index (FC = 1.03, $P = 1.81e-107$) was detected, indicating DHX36 loss induces global mRNA structurization thus decreased accessibility. Calculation of the Gini index of each of the 20 bins of the mRNAs and ΔGini (the Gini index in the KO subtracted by the WT index) showed increasing differences toward the 3′UTR regions (Fig. 1F). Consistently, further examination of the structural changes by binned reactivity profiles across the length of each region and ΔReactivity (the reactivity scores in the KO subtracted by the WT scores) revealed decreased SHAPE reactivity and increased accessibility in all regions upon DHX36 loss and 3′UTRs exhibited the largest gain of accessibility (Fig. 1G). In sum, in vivo RNA structurome mapping provides a global view of DHX36-mediated RNA structural changes.

### DHX36 binding induces RNA structural loss and gains accessibility

To elucidate the structural changes directly induced by DHX36 binding, we next interrogated DHX36-bound mRNAs (Fig. 2A). The publicly available DHX36 photoactivatable-ribonucleoside-enhanced crosslinking and immunoprecipitation (PAR-CLIP) data from HEK293

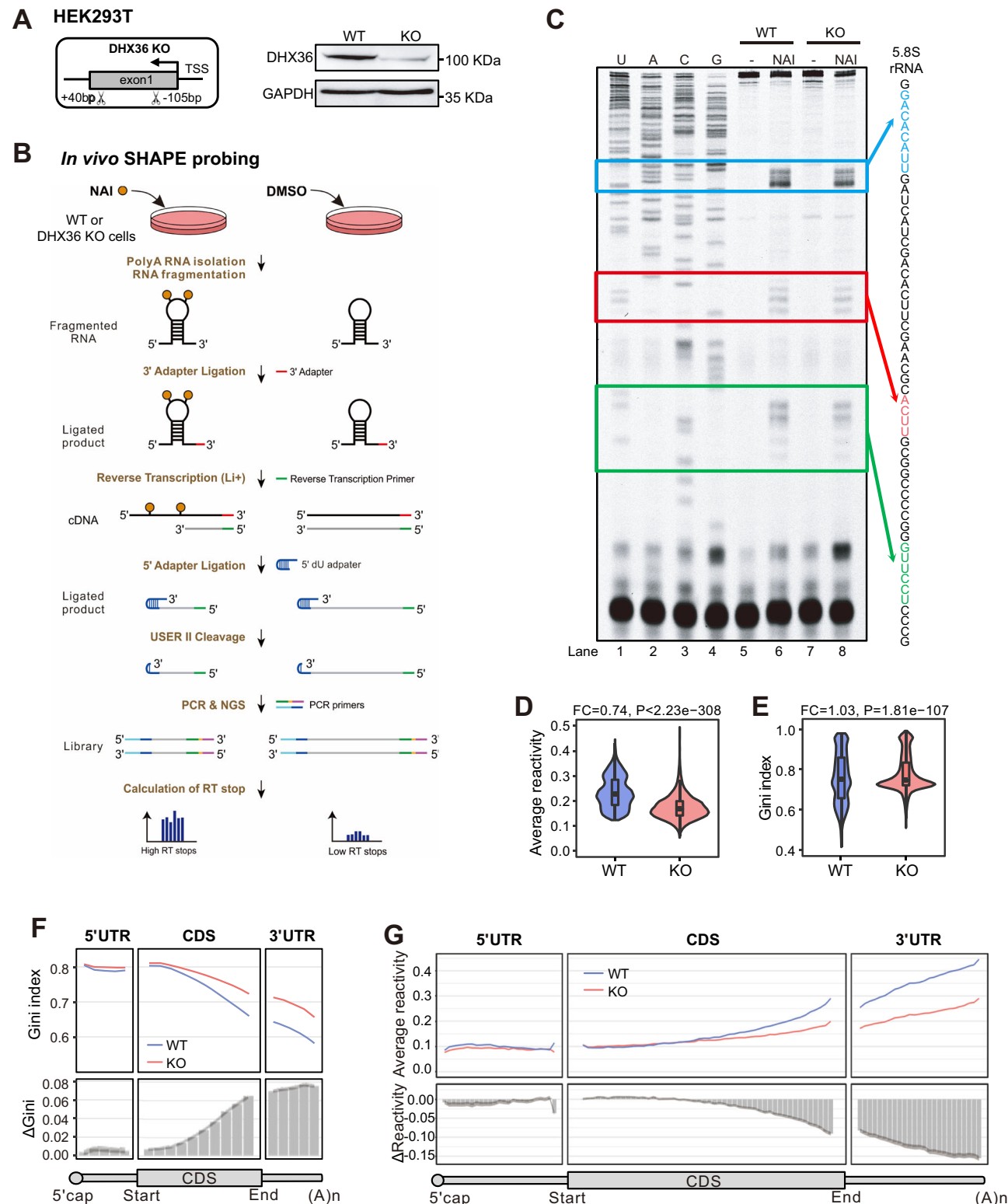

cells[37] were re-analyzed to identify crosslinking sites at single-nucleotide resolution using the CLIP Tool Kit (CTK)[38]. A total of 4361 crosslinking sites were identified (Supplementary Data 2), of which 4263 were annotated to 2886 mRNAs. Close examination revealed that DHX36 binding preferentially resided in 3′UTR regions, especially around the stop codons (Fig. 2B, C). However, 5′UTRs harbored the highest DHX36 binding density after normalization by the region length (Fig. S2A). Further prediction showed that 79% of the DHX36 binding sites possessed potential rG4s of various subtypes with the

highest enrichment and stability found at the 5′UTRs (Fig. 2D, Fig. S2B, C, and Supplementary Data 2), confirming that indeed rG4 structure is the predominant RNA substrate for DHX36 binding[22]. This was further validated by the results from in vitro RBNS in which we incubated the His-tagged RHAU53 peptide of DHX36 protein with a 40-nt random RNA library in the presence of either K+, which stabilizes rG4 folding, or Li+, which destabilizes rG4 folding. As expected, the RHAU53-bound RNAs were biased towards G-repeat 6mer motifs (Fig. S2D) and were enriched for potential rG4s (Fig. S2E) in the

**Fig. 1 | In vivo RNA structurome profiling unveils DHX36 depletion-induced global increase of RNA structures. A** Left: schematic illustration of the CRISPR-Cas9 mediated generation of DHX36 knockout (KO) in HEK293T cells. 169 bp of exon 1 of the DHX36 gene was deleted by two single-guide RNAs and CRISPR-Cas9. Right: western blot confirmed the inactivation of DHX36 in KO cells with GAPDH as the loading control. Source data are provided as a Source Data file. **B** Schematic illustration of in vivo structure mapping in WT and DHX36-KO cells. Orange circle, SHAPE modification; NGS next-generation sequencing, RT reverse transcription. **C** Sequencing gel for 5.8S rRNA showing the RT stops induced by NAI modification. Source data are provided as a Source Data file. **D** The average SHAPE reactivity score and **E** Gini index of reactivity scores of all mRNAs (*n* = 7792) in WT and

DHX36-KO. The most abundant mRNA per gene was analyzed. Wilcoxon signed-rank test was used to calculate the statistical significance. The average fold change (FC) between the KO vs WT is shown. The boxes indicate median (center), Q25, and Q75 (bounds of box), the smallest value within 1.5 times interquatile range below Q25 and the largest value within 1.5 times interquatile range above Q75 (whiskers). **F** Top: the binned average Gini index across the length of 5′UTRs (5 bins), CDSs (10 bins), and 3′UTRs (5 bins) of all mRNAs. Bottom: The binned ΔGini (KO-WT) across the above regions. **G** The binned average reactivity and ΔReactivity (KO-WT) across the 5′UTR (25 bins), CDS (50 bins) and 3′UTR (25 bins). The shaded area represents 95% confidence intervals (CIs) of the average ΔGini or ΔReactivity of each bin calculated by paired two-sided Student's *t*-test.

presence of K⁺. The highest enrichment of K⁺ RBNS motifs was detected in 5′UTR binding sites supporting the preferred binding of DHX36 to rG4s in 5′UTRs (Fig. S2F).

As 202 DHX36-bound mRNAs harbored the binding sites in both 5′UTR and 3′UTR, we examined whether this might be a consequence of the presence of long-range interactions between these regions that were captured during CLIP. By analyzing the publicly available PARIS (Psoralen Analysis of RNA Interactions and Structures) data in HEK293T cells[39], we found that DHX36-bound mRNAs were indeed enriched for inter-regional long-range RNA-RNA interactions (1893 out of 2886, 65.60%) (Fig. S2G). Additionally, 18.75% (541 out of 2886) of the DHX36-bound mRNAs harbored DHX36 binding sites overlapped with at least one arm (one interactive RNA fragment in PARIS), which is significantly higher than the randomly shuffled sites (Fig. S2H). These results indicate that DHX36 binding may induce long-range structural rearrangement. Further analysis identified 25 DHX36-bound mRNAs with binding sites overlapped with both arms of long-range interacting sites, and only 1 harbored 5′UTR-3′UTR interaction (Fig. S2I), indicating the simultaneous binding of DHX36 on 5′UTR and 3′UTR regions of the same mRNA may not be a consequence of long-range interactions.

Next, our analysis uncovered that the above-defined DHX36-bound mRNAs showed significantly decreased reactivity scores (FC = 0.68, *P* = 1.66e-238) and increased Gini index (FC = 1.06, *P* = 7.14e-80) in KO cells compared to WT (Fig. 2E), suggesting DHX36 binding induces mRNA structural loss thus increases structural accessibility. Consistently, compared to unbound mRNAs, DHX36-bound mRNAs displayed a larger loss in reactivity (*P* = 4.56e-53) and gain in Gini index (*P* = 5.22e-28) in KO vs. WT cells (Fig. 2F). Further analyses of each transcript region revealed that 3′UTRs exhibited the largest structural gain upon DHX36 loss, followed by CDSs and 5′UTRs (Fig. 2G, H and Fig. S3A); DHX36-bound mRNAs also showed the largest structural changes in 3′UTR, compared to the unbound mRNAs (Fig. S3B). To investigate whether the structural gain towards the 3′end is driven by innate sequence features, we correlated the regional reactivity change with GC-content or sequence length of the designated region and found the large structural change of 3′UTRs was probably caused by low GC content (*r* = 0.312, *P* = 8.40e-113; Fig. 2I and Fig. S3C–E).

Since the above comparison of average SHAPE reactivity or Gini index cannot pinpoint the exact structural changing sites across the entire mRNA, we next employed dStruct[40] to uncover differentially reactive regions (DRRs). As a result, a total of 50,683 DRRs were identified from 2419 DHX36-bound mRNAs; and only 3564 possessed DHX36 binding sites (Fig. 2J and Supplementary Data 3), indicating DHX36 binding leads to structural alterations at both localized binding sites and also distal sites without direct binding. Furthermore, DHX36 loss mainly led to a structural gain of DRRs, with 74% of DRRs exhibiting decreased average SHAPE reactivity while 26% increased (Fig. S3F). These DRRs were most significantly enriched for the G-rich sequence motif (Fig. S3G) and were differentially enriched for the paired structural motifs in the KO vs. WT (Fig. S3H). Metagene analysis revealed that DRRs preferentially resided in 3′UTRs, followed by CDSs and 5′UTRs (Fig. 2K). Upon normalization by the number of regional

binding events, we found that 3′UTRs were most enriched for DRRs (Fig. S3I), indicating that DRRs were not evenly distributed among the binding sites.

The discrepancy between the preferential binding of DHX36 to 5′UTRs (Fig. S2A) and the enrichment of DRRs within 3′UTRs (Fig. S3I) led us to speculate that DHX36 binding in different regions may not confer the same structural modifying effect. To test this notion, we selected the mRNAs with exclusive DHX36 binding in only one region, 3′UTRs or CDSs or 5′UTRs to compare the DRR distribution. We found that DRRs were enriched towards the end of CDSs or 3′UTRs with limited dependence on DHX36 binding location (Fig. 2L). Consistently, binding events located in any transcript region induced higher changes of SHAPE reactivity and Gini index in 3′UTRs but exerted limited effect on 5′UTRs (Fig. S3J–L). For example, DHX36 binding to the 5′UTR of mRNA *PYCR1* and the 3′UTR of *CCNA2* both led to the highest reactivity increase and structural loss in their 3′UTRs (Fig. 2M).

## DHX36 induces a localized structural loss in 3′UTRs binding sites

After dissecting the structural changes at whole-transcript and regional scale, we further examined localized structural effects surrounding DHX36 binding sites (±50 nt of the crosslink site, which showed the largest structural changes compared to longer regions (Fig. S4A)). As expected, SHAPE reactivity at the DHX36 binding sites dramatically decreased in KO vs. WT cells (Fig. 3A), supporting that DHX36 loss induces local structural gain and reduces accessibility. Given that SHAPE reactivity only provides probabilistic inference of RNA secondary structures but cannot be used to distinguish paired from single-stranded bases, we then calculated the base-pairing probability (BPP) constrained by SHAPE reactivity. Consistently, higher BPPs were observed in KO cells, further confirming the reduced structural accessibility upon DHX36 loss (Fig. 3A). Next, we examined the DRR distribution within DHX36 binding sites and found 71.69% of the binding sites possessed DRRs (Fig. 3B and Supplementary Data 3). Five bins were generated for each DRR, and lower reactivity was detected in KO cells (Fig. S4B), indicating that most DRRs gained in structures upon DHX36 loss. Altogether, the above results suggest that DHX36 binding induces structural loss at the binding site, which is in line with its known helicase unwinding function. Interestingly, closer examination uncovered similar proportions and subtypes of rG4s between DRR-containing (Fig. 3C, 78%) and all DHX36 binding sites (Fig. 2D, 79%), suggesting that DHX36 did not discriminately unwind rG4s; this was also supported by similar distribution of reactivity scores and BPPs between rG4-containing and non-rG4 sites (Fig. 3D and Fig. S4C).

To further interrogate localized structural changes at different mRNA regions, we classified the DHX36 binding sites by regions and separately profiled reactivity and BPPs. Of note, the largest decrease in SHAPE reactivity (Fig. 3E) and increase in BPP (Fig. S4D) upon DHX36 loss was observed in the 3′UTR binding sites. Moreover, more than half of DRRs were located within 3′UTR regions (Fig. 3F). Furthermore, after normalization by the total number of binding sites in each region, DRRs were most enriched in the 3′UTR binding sites (Fig. S4E),

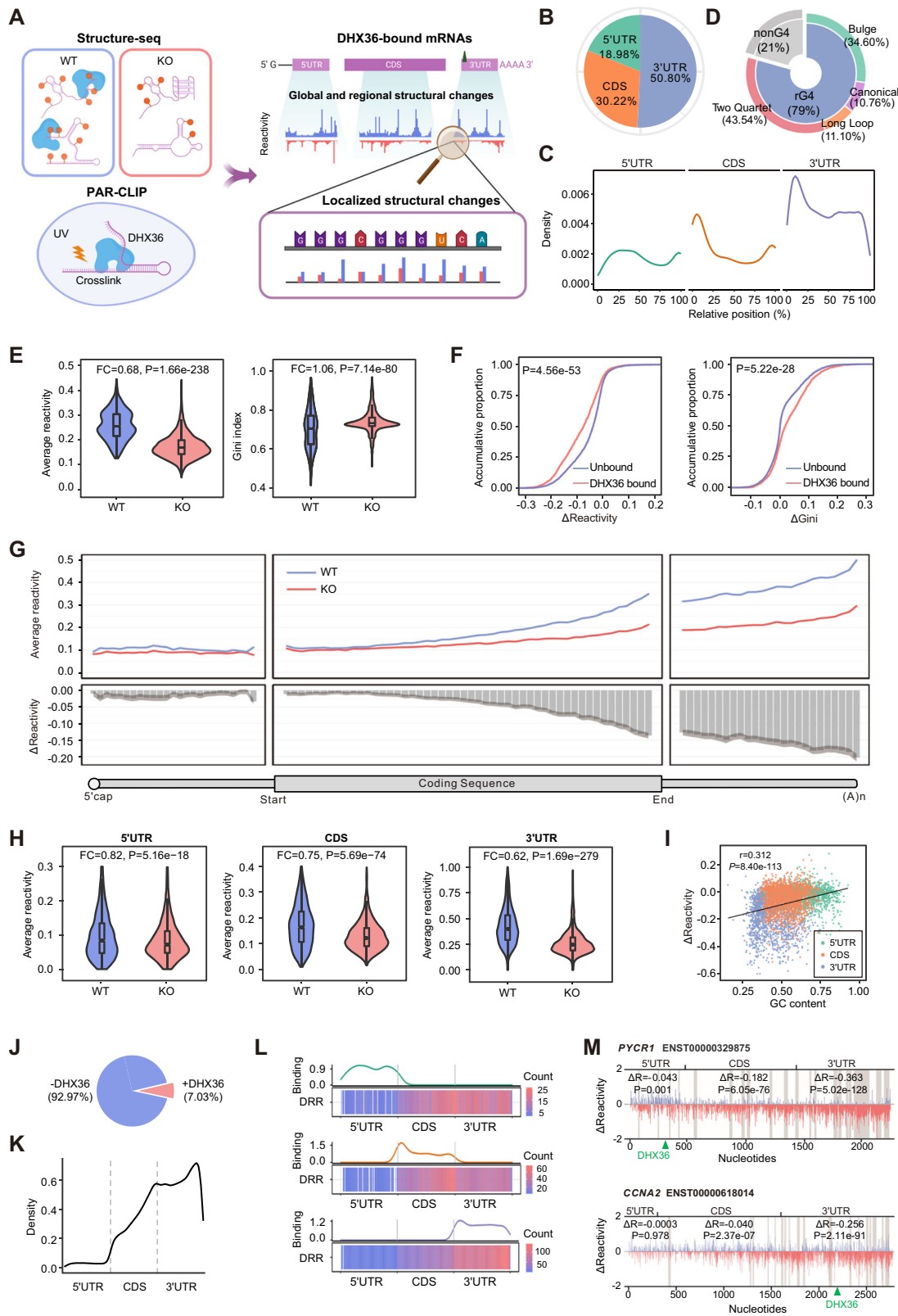

indicating 3'UTRs undergo the largest structural change upon DHX36 binding. The structure folding predicted by Vienna RNA package[41] using SHAPE reactivity as a constraint further illustrated extensive structure rearrangement in the 3'UTRs. For example, DHX36 loss resulted in a less accessible structure at the binding site within the 3' UTR of *JUN* mRNA (Fig. 3G, H). In addition, to examine the specificity of DHX36 binding on the localized mRNA structural changes, we

compared the reactivity change between the DHX36 binding sites and the randomly selected sites of the designated region of the same mRNA. As shown in Fig. S4F–H, 5'UTR and CDS but not 3'UTR binding sites showed significantly higher structural changes compared to random sites.

Next, to investigate why 3'UTR binding sites harbored the highest structural increase upon DHX36 loss, we speculated that it

**Fig. 2 | DHX36 binding induces mRNA structural loss and gained accessibility.**
**A** The workflow for analyzing PAR-CLIP and Structure-seq data to identify DHX36 binding-induced structure remodeling. Created in BioRender. Zhang, Y. (2022) BioRender.com/p72o682. **B** The distribution of DHX36 binding sites in 5′UTRs, CDSs, and 3′UTRs. **C** The transcriptomic distribution of DHX36 binding sites. **D** rG4 formation was predicted in DHX36 binding sites, and the distribution of each subtype is shown. **E** The average SHAPE reactivity and Gini index of the reactivity scores of DHX36-bound mRNAs ($n = 1787$). The most abundant mRNA per gene was analyzed. The significance was calculated by a two-sided Wilcoxon signed-rank test. The fold change (FC) between the KO vs WT is shown. **F** Comparison of ΔReactivity and ΔGini (KO-WT) between DHX36-bound and unbound mRNAs. The significance was calculated by a two-sided Wilcoxon rank-sum test. **G** The binned average reactivity (top) and ΔReactivity (bottom) across the length of 5′UTRs (25 bins), CDSs (50 bins), and 3′UTRs (25 bins) of DHX36-bound mRNAs. The shaded area was plotted in the same way as Fig. 1G. **H** The average reactivity of 5′UTR, CDS, and 3′UTR of DHX36-bound mRNAs ($n = 1787$). The significance was calculated by a two-

sided Wilcoxon signed-rank test. The boxes indicate median (center), Q25 and Q75 (bounds of box), the smallest value within 1.5 times interquartile range below Q25 and largest value within 1.5 times interquartile range above Q75 (whiskers). **I** Scatterplot showing regional ΔReactivity vs. GC content of the designated regions. r and P represent the correlation coefficient and statistical significance calculated by the Pearson correlation test. **J** The proportion of DHX36 depletion-induced DRRs located within (+DHX36) and outside (−DHX36) DHX36 binding sites. **K** The distribution of DHX36-induced DRRs across different regions of DHX36-bound mRNAs. **L** The distribution of DRRs across the mRNAs with DHX36 binding to 5′UTR (top), CDS (middle) or 3′UTR (bottom) only. The line plot shows the location of DHX36 binding sites, and the heatmap shows the DRR enrichment. **M** ΔReactivity (KO-WT) of *PYCR1* and *CCNA2* mRNA. DRRs and DHX36 binding sites are high-lighted in the shaded areas and green arrows, respectively. ΔR and P represent the average difference in reactivity and the significance calculated by the two-sided Wilcoxon signed-rank test.

may be related to high GC contents since GC base pairs are more stable than AU base pairs and thus more resistant to DHX36 unwinding[42]. Alternatively, this could be caused by the relatively high average sequence length of 3′UTRs compared to 5′UTRs and CDSs (Fig. S3D)[43,44]. To test this, we established multivariable linear regression models, in which the localized reactivity changes of all DHX36 binding sites or those located in 5′UTRs, CDSs, or 3′UTRs were each used as the response variable. The GC content of the DHX36-bound region (Fig. S3C), the localized GC content of the 100nt DHX36 binding site (Fig. S4I), and the length of the bound region were not collinear to each other (Supplementary Data 4), thus used as the explanatory variables. In the model based on all DHX36 binding sites (Fig. 3I, top row), we found that low GC contents of the entire transcript regions (regional GC contents), where DHX36 binding sites reside, to a large extent explained the highest localized structural changes of 3′UTR binding sites, while the GC contents of localized binding sites (localized GC contents) and the length of the transcript regions conferred limited contribution. Notably, the above phenomenon was not observed in the models for binding sites in individual mRNA regions (the last three rows of Fig. 3I), suggesting that differential regional GC contents impact the unwinding effect of DHX36. In addition, the limited contribution of localized GC contents to structural changes indicated that DHX36 binding exerted dis-criminative unwinding power in the susceptible contexts (i.e., 3′UTRs with low regional GC contents).

## DHX36 binding-induced structurome remodeling effect is con-served in C2C12 myoblast cells

To examine whether the effect of DHX36 binding on RNA structure remodeling is conserved in other cell types, we next conducted Structure-seq with DMS in our previously established WT and Dhx36-KO C2C12 mouse myoblast cell lines[22]. The good correlation of the identified RT stops between the two replicates and the concordance between the reactivity score, as well as the phylogenetically derived 18S rRNA structures, demonstrated the high quality of our data (Fig. S5). As a result, a total of 25,637 mRNAs were identified with sufficient RT coverage (Supplementary Data 5). Integrating our published DHX36 CLIP-seq from C2C12 cells[22], a similar analysis was performed to define DHX36 binding-induced RNA structural changes. DMS reactivity indicated that DHX36 loss induced significant structural loss in DHX36-bound mRNAs, especially at 3′UTRs (Fig. S6A, B), resembling the findings from HEK293T cells (Fig. 2G, H). Next, we took a closer view of the localized structural changes at different mRNA regions and also detected the most pronounced changes at the DHX36-binding sites within 3′UTRs (Fig. S6C). Altogether, the above findings demonstrate that DHX36 binding-induced structural loss may be a conserved phe-nomenon in various cells, which triggered us to further elucidate the functional consequences.

## DHX36 induced 3′UTR structural change is associated with post-transcriptional regulation of mRNA abundance

Secondary structural change within 3′UTRs is an emerging mechanism in post-transcriptional regulation of mRNA abundance via modulating RNA stability or degradation[12,45–47]. To examine whether the DHX36 induced 3′UTR structural changes are associated with RNA abundance, we took advantage of the existing RNA-seq data generated from whole-cell RNAs or chromatin-associated RNAs in WT or DHX36-KO HEK293 cells[37] and calculated the RNA abundance fold change (FC) in whole cell and chromatin fraction respectively; a post-transcriptional FC (pFC) index was then deduced by dividing the chromatin by the whole cell FC (Fig. 4A). With a threshold of pFC >1.5, 1998 genes were iden-tified as post-transcriptionally downregulated (pDGs) upon DHX36 loss and 1850 upregulated genes (pUGs) (Fig. 4B). Furthermore, 149 (7.5%) pDGs and 517 (27.9%) pUGs bound by DHX36 were defined as DHX36 post-transcriptional regulatory targets ($P = 6.43e-63$; Fig. 4C and Supplementary Data 6). These target genes displayed higher pFCs compared to all protein-coding genes, ($P = 2.58e-36$, Fig. 4D), indicat-ing that DHX36 loss leads to a post-transcriptional increase and DHX36 binding is indeed associated with post-transcriptional regulation of mRNA abundance. Next, we correlated the structural changes of the target mRNAs with the abundance changes and uncovered that the pUG targets showed higher reactivity loss thus structural gain com-pared with the pDGs ($P = 6.04e-10$; Fig. 4E), suggesting that DHX36 binding-induced structural loss is associated with post-transcriptional downregulation of mRNA abundance. Next, we calculated the corre-lation of the fold change of average reactivity of the entire transcript with pFCs and found that reactivity changes were inversely correlated with RNA abundance changes (Fig. 4F). When each transcript region was examined, an inverse correlation was also observed (Fig. 4G-I); compared to 5′UTRs and CDSs which showed modest correlation ($r = −0.05$, $P = 0.001$ for 5′UTRs; $r = −0.06$, $P = 1.35e-06$ for CDSs; Fig. 4G, H), a higher correlation was observed between the structural gain in 3′UTRs and the mRNA abundance increase ($r = −0.18$, $P = 2.75e-39$; Fig. 4I), suggesting that 3′UTR structures may exert dominant contribution in post-transcriptional regulation of mRNA abundance. Moreover, hierarchical clustering identified subsets of mRNA that were strongly affected by the DHX36-induced structural changes (Fig. 4J and Supplementary Data 7). For example, cluster 2 in the 5′ UTR group showed a strong negative correlation between structural changes and mRNA abundance changes (Fig. 4J, left), whereas cluster 5 in the 3′UTR group showed a positive correlation (Fig. 4J, right). Interestingly, GO analysis indicated that these mRNAs were enriched for diversified functional processes with several clustered enriched for RNA processing-related functions such as RNA splicing, RNA metabolic process, ncRNA processing et al (Fig. S7A–C).

In addition, we compared the localized structural changes of pUG and pDG targets and found that pUGs showed higher reactivity loss,

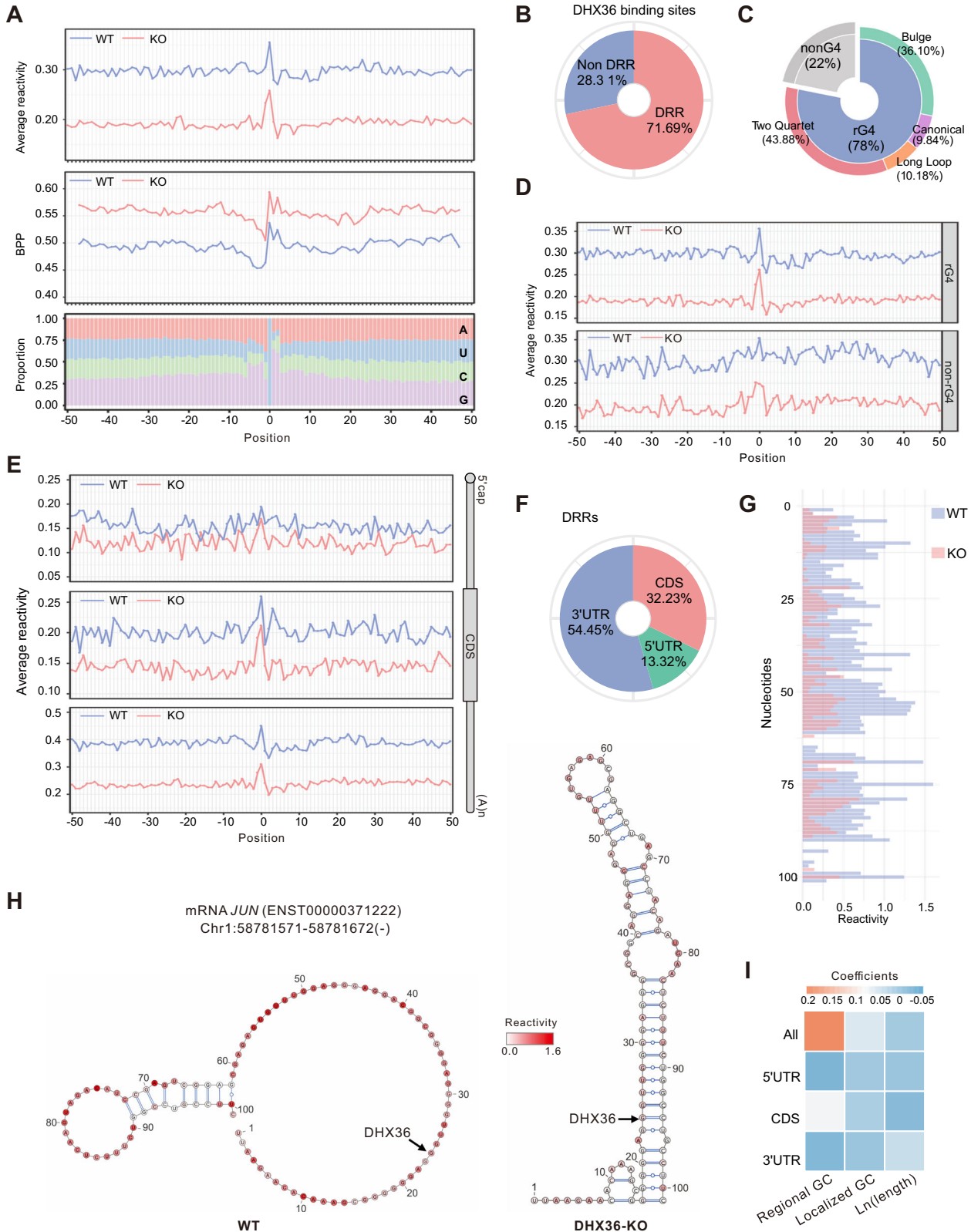

thus structural gain than pDGs upon DHX36 inactivation (Fig. 4K). Moreover, the most overwhelming localized structural increase of pUGs was observed in 3′UTR binding sites (Fig. 4L). Altogether the above findings suggest that DHX36 binding-induced 3′UTR structural decrease is associated with post-transcriptional downregulation of mRNA abundance.

## DHX36-induced RNA structural accessibility facilitates m⁶A-dependent YTDHF1 binding and RNA degradation

To further fathom the underlying mechanism of how DHX36 binding regulates mRNA abundance via structural modulation, we realized that the enrichment of DHX36 binding sites within 3′UTRs especially around stop codons, was reminiscent of the distribution of m⁶A

**Fig. 3 | DHX36 induces localized structural loss of 3'UTR binding sites.**
**A** Average SHAPE reactivity (top), BPP (middle), and nucleotide composition
(bottom) across DHX36 binding sites. Position 0 represents the exact crosslink
site identified by PAR-CLIP data. **B** The percentage of DHX36 binding sites with or
without DRRs. **C** rG4 formation was predicted in DRR-containing DHX36 binding
sites, and the distribution of each subtype is shown. **D** Comparison of average
SHAPE reactivity across DHX36 binding sites with and without rG4s in WT and
DHX36-KO. **E** Comparison of average SHAPE reactivity across DHX36 binding sites
located in 5'UTR (top), CDS (middle), and 3'UTR (bottom) in WT and DHX36-KO.
**F** The percentage of DHX36-bound DRRs in 5'UTR, CDS, and 3'UTR. **G** Barplot

showing the SHAPE reactivity of the DHX36 binding site within mRNA *JUN* in WT
and DHX36-KO. **H** Illustration of the folded structure of the DHX36 binding site
within mRNA *JUN* in WT and DHX36-KO. Nucleotides were color-coded based on
the SHAPE reactivity scores. The black arrow indicates the exact crosslink site.
**I** Heatmap depicting the coefficients of the explanatory variables in the linear
regression models, including localized GC content of DHX36 binding sites, regional
GC content, and ln(length) of DHX36-bound mRNA regions. Rows represent the
models designed for four kinds of response variables: the localized ΔReactivity
(KO-WT) of all, 5'UTR, CDS, and 3'UTR binding sites.

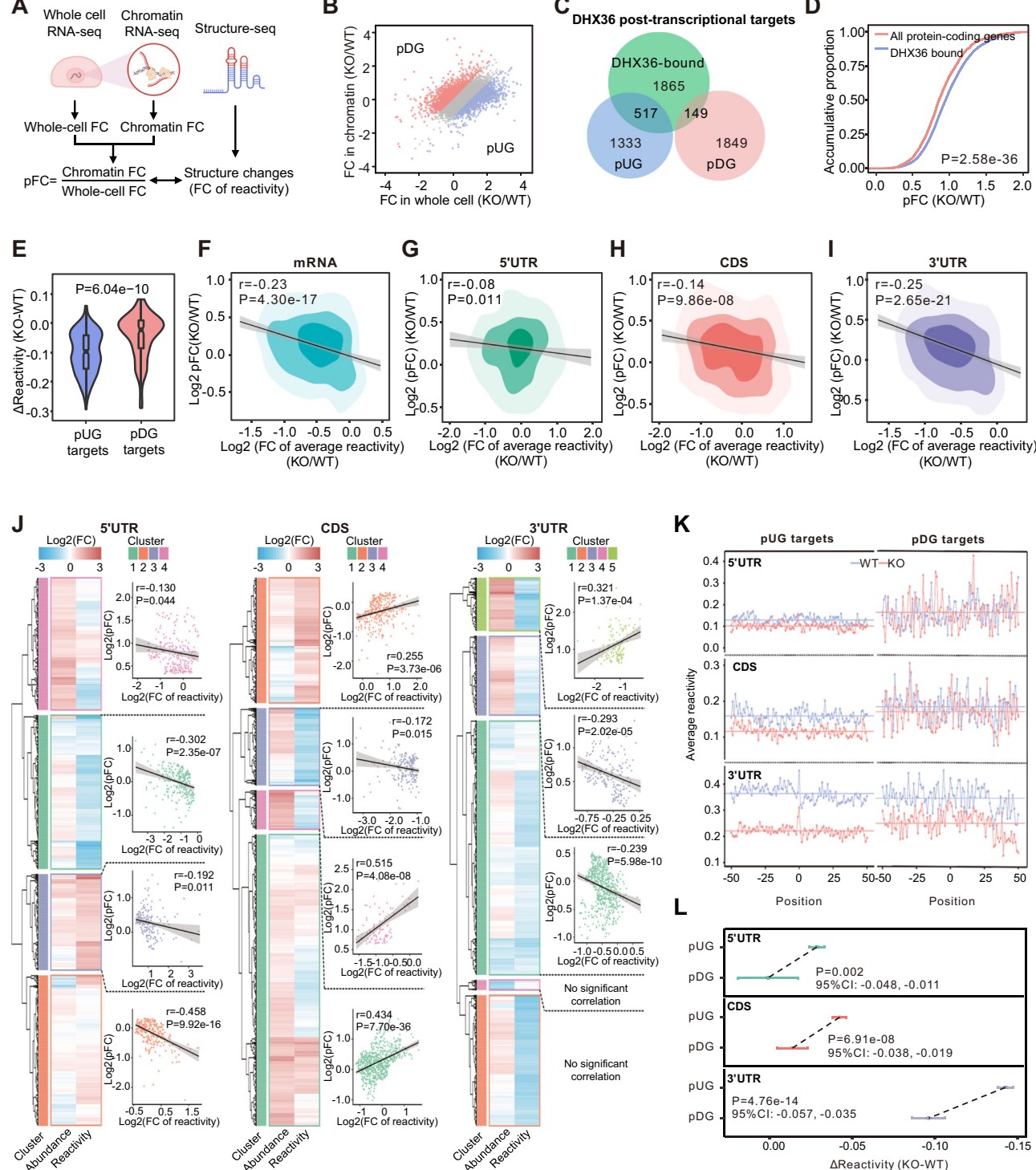

**Fig. 4 | DHX36 binding-induced 3'UTR structural change is associated with post-transcriptional regulation of mRNA abundance. A** Schematic illustration showing the method for defining DHX36-induced post-transcriptional regulatory effect. The post-transcriptional fold change (pFC) of mRNA abundance was calculated by dividing the fold change in chromatin fraction by that in the whole cell. Created in BioRender. Zhang, Y. (2021) BioRender.com/a92l088. **B** Scatterplot showing the definition of pUGs and pDGs based on the pFC of mRNA abundance. **C** Venn diagram showing the overlapping between DHX36-bound mRNAs and pUGs or pDGs. Statistical significance was calculated by the Chi-square test. **D** Comparison of pFCs of DHX36-bound mRNAs and all protein-coding genes. **E** Comparison of ΔReactivity of pDG ($n = 149$) and pUG targets ($n = 517$). The boxes indicate median (center), Q25 and Q75 (bounds of box), the smallest value within 1.5 times interquatile range below Q25 and largest value within 1.5 times interquatile range above Q75 (whiskers). A two-sided Wilcoxon rank-sum test was used to calculate the statistical significance in (**D**, **E**). **F**−**I** Scatterplots showing the correlations of mRNA abundance pFCs with reactivity changes of (**F**) entire mRNAs, **G** 5'UTRs, **H** CDSs, and **I** 3'UTRs of DHX36-bound mRNAs. r and P represent the correlation coefficient and statistical significance calculated by the Pearson correlation test. **J** Heatmaps showing the hierarchical clustering on the DHX36-bound mRNAs based on their abundance changes and the average reactivity changes of 5'UTR (left), CDS (mid), and 3'UTR (right). For each cluster, the Pearson correlation between the structure changes and the abundance changes was displayed in a scatterplot. The error bands in (**F**−**J**) represent the 95% CIs. **K** Average SHAPE reactivity across DHX36 binding sites within the designated regions of pUGs and pDGs. The colored horizontal lines represent the means of average reactivity across the binding sites. Position 0 represents the exact crosslink sites identified by PAR-CLIP data. **L** ΔReactivity of DHX36 binding sites within the designated regions of pUGs ($n = 517$) and pDGs ($n = 149$). Error bars represent 95% CI of the mean of ΔReactivity. P values were calculated by unpaired two-sided Student's t-test.

modification[48] and YTHDF1 binding[49]. Moreover, it has been reported that potential rG4s co-localize with m6A sites[50,51] and YTHDF1 can regulate mRNA degradation in HEK293 cells[25,26], triggering us to speculate that the DHX36 effect on mRNA abundance may be mediated by m6A/YTHDF1 (Fig. 5A). To test the above hypothesis, we leveraged the available miCLIP data[48] and iCLIP data for YTHDF1[49] to identify m6A and YTHDF1 binding within the DHX36 3'UTR binding sites. As a result, these sites were significantly enriched for canonical m6A and YTHDF1 binding motifs (Fig. 5B and Fig. S8A). Furthermore, ~20% of these 3'UTR DHX36 binding sites overlapped with m6A and 24% with YTHDF1 binding, which significantly overwhelmed the overlapping with randomly selected sites ($P < 0.01$, Fig. 5C, D and Supplementary Data 8). The above results demonstrate that DHX36 binding co-localizes with m6A/YTHDF1 within 3UTRs.

Next, to investigate whether the co-binding is associated with DHX36 regulation of post-transcriptional RNA abundance, we found DHX36-bound mRNAs with m6A/YTHDF1 within their 3'UTRs showed significantly higher pFCs, compared to those without m6A/YTHDF1 ($P = 5.46e\text{-}22$, Fig. 5E). To further elucidate whether YTHDF1 regulates RNA degradation, we harnessed the available 4-thiouridine labeling (4SU-TT-seq) data generated from WT and YTHDF1-KO HEK293 cells[25]. 4SU-TT-seq infers mRNA half-life by estimating the fraction of 4SU-labeled nascent RNAs in the total RNA pool. A total of 3738 mRNAs showed prolonged half-life (KO/WT >1) upon YTHDF1 loss, thus defined as YTHDF1 degradation targets. Among the 327 pUG targets with DHX36 binding at their 3'UTRs, 157 were significantly overlapped with these YTHDF1 degradation targets and defined as DHX36/m6A/YTHDF1 degradation targets ($P = 0.004$, Fig. 5F); but no significant overlapping was observed for pDG targets ($P = 0.445$, Fig. S8B). The above results suggest that YTHDF1 co-binding with DHX36 leads to mRNA degradation. Furthermore, significant overlapping between 3'UTR YTHDF1-binding sites and DRR-containing DHX36 binding sites within 3'UTRs ($P = 1.81e\text{-}12$, Fig. 5G) was also detected, suggesting a possibility that DHX36 binding-induced 3'UTR structural accessibility may facilitate YTHDF1 binding. As YTHDF1 binding activity can be influenced by both m6A level[52] and substrate structure[33], we tested if DHX36 loss leads to reduced m6A level which was inferred by probing the structural signature at the m6A site according to the m6A switch theory[3]. The m6A-induced signature structural decrease was observed in both WT and DHX36-KO cells (Fig. S8C). Moreover, a similar structural decrease pattern at the m6A sites located within DHX36 binding sites was detected in both WT and DHX36-KO cells (Fig. 5H), demonstrating that DHX36 inactivation exerted limited influence on the m6A level. This leads us to speculate that DHX36 binding-induced structural accessibility may expose m6A sites to facilitate YTHDF1 binding and subsequent mRNA degradation. To substantiate the notion, YTHDF1 CLIP-seq was performed in WT and DHX36-KO cells. As expected, the identified YTHDF1 binding sites were enriched for m6A motifs in both cells (Fig. 5I and Fig. S8D), supporting that DHX36 loss probably did not have an impact on the m6A level. However, a significantly decreased YTHDF1 binding peak number (Fig. 5J) and attenuated binding signals (Fig. 5K) were identified in the DHX36-KO vs. WT cells.

## DHX36 facilitates YTHDF1 binding to decrease mRNA stability on selected targets

To further test the above notion, we selected four representative mRNAs including EZR (ENST00000367075), MEPCE (ENST00000310512), PHF23 (ENST00000320316), and ZNF768 (ENST00000380412) from the DHX36/m6A/YTHDF1 degradation target list and performed additional experimental validation in HEK293T cells. The abundance of these mRNAs was increased in the whole cell, but not chromatin faction upon DHX36 loss (Fig. 6A). A DHX36 binding site was identified within the 3'UTR of each target, and this site was also co-bound by m6A/YTHDF1. The SHAPE reactivity and structural accessibility of these sites showed a significant decrease upon DHX36 inactivation (Fig. 6B). Treatment of the cells with STM2457 that inhibits METTL3 activity to decrease m6A level (Fig. S7E), did not reduce significant structural accessibility on these mRNAs (Fig. S7F−I), suggesting that DHX36-induced structural changes may not be induced by m6A. Next, the decreased levels of these mRNAs in DHX36-KO vs. WT were confirmed by RT-qPCR (Fig. 6C). When the cells were treated with actinomycin D for 4 or 8 h, all four mRNAs showed increased RNA stability and decreased degradation rate in the DHX36-KO vs. WT cells (Fig. 6D), confirming that DHX36 regulates the degradation of these mRNAs. Additionally, EGFP reporters were generated by fusing the DHX36 3'UTR binding site of each mRNA downstream of an EGFP reporter (Fig. 6E). Expectedly, decreased degradation rate and increased stability of the EGFP mRNA were observed in the DHX36-KO vs. WT cells (Fig. 6F). Furthermore, the calculated half-life of these mRNAs with the 4SU-TT-seq data[53] were significantly increased upon YTHDF1 inactivation (Fig. 6G). To further examine whether YTHDF1 binding to m6A was facilitated by DHX36 binding, we conducted YTHDF1-RNA immunoprecipitation (RIP) (Fig. 6H) and found YTHDF1 binding to all four mRNAs was significantly impaired in DHX36-KO vs. WT cells (Fig. 6I); and the decreased interaction was not due to reduced YTHDF1 expression (Fig. 6J). Altogether, the above results validate that DHX36 binding-induced structural accessibility in 3'UTRs enhances m6A-dependent YTHDF1 binding to promote target mRNA degradation (Fig. 7).

## Discussion

In this study, we elucidated DHX36 orchestrated RNA structural remodeling and how it controls post-transcriptional RNA abundance. Harnessing in vivo RNA Structure-seq, we probed transcriptomic structural changes induced by DHX36 loss. Combining with DHX36 binding profiles, we uncovered that DHX36 binding induces structural loss, thus increasing RNA accessibility, especially

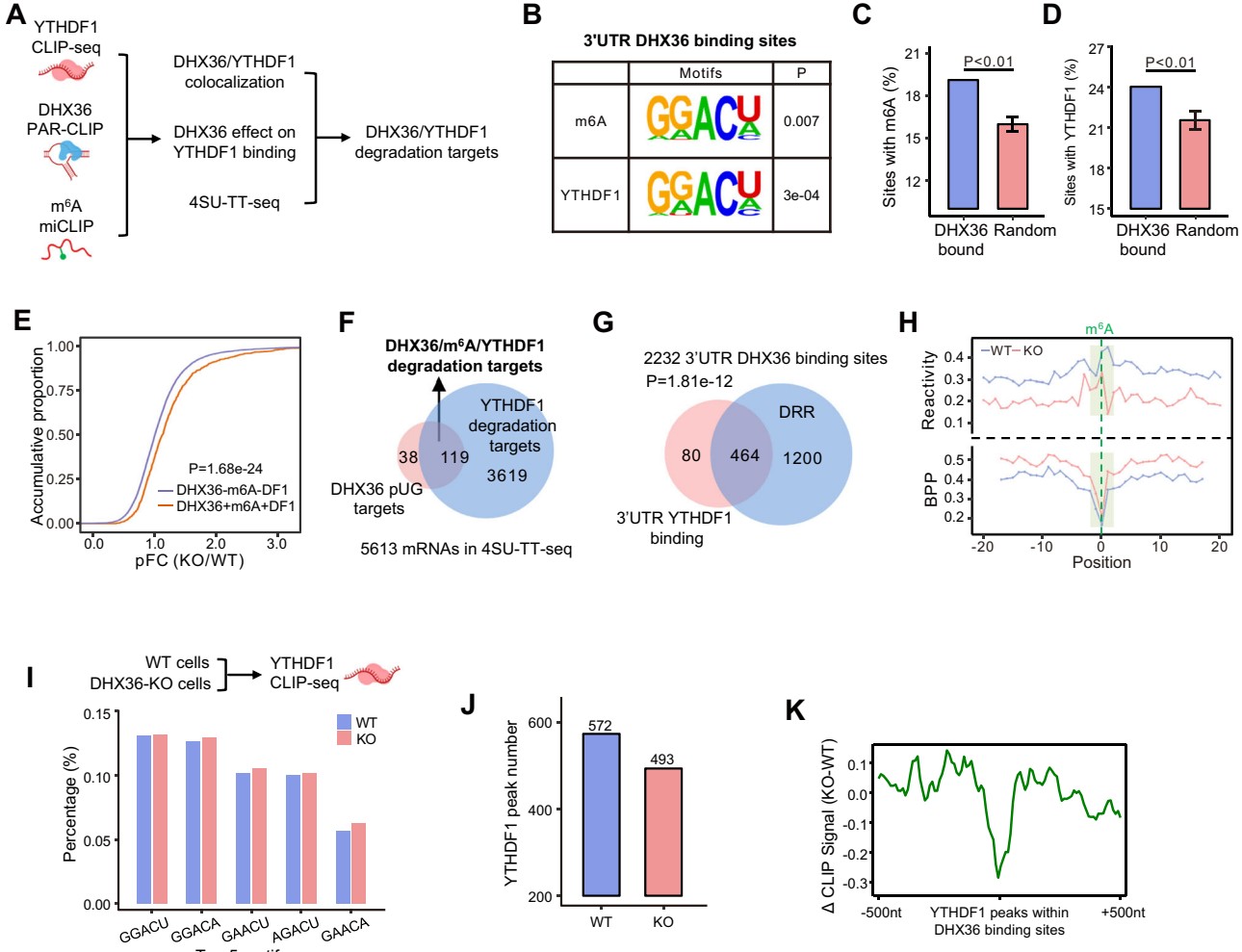

**Fig. 5 | DHX36-induced RNA structural accessibility facilitates m⁶A-dependent YTDHF1 binding and RNA degradation. A** Schematic illustration showing the investigation of DHX36/m⁶A/YTHDF1 co-regulation of mRNA abundance via RNA structure remodeling. Created in BioRender. Zhang, Y. (2022) BioRender.com/r15e384. **B** Significant enrichment of m⁶A and YTHDF1 motifs in the 3′UTR binding sites of DHX36. **C, D** Enrichment of m⁶A and YTHDF1 binding within the true set ($n = 1$) of DHX36 binding sites compared to 100 sets ($n = 100$) of random sites. The randomly selected sites in each set are in equal numbers, equal length, and from the same 3′UTR with the DHX36 binding sites. Data are presented as mean values ± SD. Statistical significance was calculated by the Monte Carlo method. **E** Comparison of pFCs between DHX36-bound mRNAs with m⁶A or m⁶A/YTHDF1 within 3′UTRs and those without m⁶A/YTHDF1. Statistical significance was calculated by the Wilcoxon

rank-sum test. **F** Identification of DHX36/m⁶A/YTHDF1 degradation targets by comparing DHX36 pUG targets with YTHDF1 degradation targets. **G** Venn diagram showing significant overlapping between 3′UTR YTHDF1 binding sites and DRR-containing DHX36 binding sites within 3′UTRs. Statistical significance in (**F**, **G**) was calculated by hypergeometric test. **H** Comparison of average reactivity and BPP of the regions surrounding m⁶A sites within DHX36-bound mRNAs in WT and DHX36-KO cells. Position 0 and green area denote m⁶A residues and m⁶A motifs, respectively. **I** Histogram showing the top five m⁶A motifs of YTHDF1 binding sites in WT and DHX36-KO HE293T cells. **J** Comparison of YTHDF1 binding peak numbers in WT and DHX36-KO cells. **K** The difference in the normalized ΔSignal (KO-WT) of YTHDF1 peaks within DHX36 binding sites.

in 3′UTR regions. Moreover, we discovered that the DHX36-induced 3′UTR structural changes are associated with post-transcriptional regulation of RNA abundance through modulating YTHDF1 binding to m⁶A sites.

Our study provides the first view of how DHX36 modulates global RNA structures. DHX36 is best known for its G4 binding and unwinding ability. Transcriptome-wide binding profiling has been reported by us and several other groups to show its predominant binding on rG4s, especially within 5′UTR regions[22,37,54]. The unwinding effect of DHX36 binding on target rG4s has also been characterized extensively[21,22,55–57]. In these studies, in vivo DHX36-induced structural changes were quantified by either indirect measurement (using rG4-specific chemical ligand or antibody) or disrupting rG4 forming sequences, but neither provides enough resolution for a closer look into DHX36 binding sites and disrupting rG4 forming sequences is also low-throughput. Moreover, despite rG4 being the dominant structure substrate for

DHX36, it can also bind with other structure elements such as RNA duplexes[15]. Therefore, the Structure-seq allowed us to interrogate in vivo transcriptomic structure remodeling induced by DHX36. The high resolution enabled us to precisely pinpoint the structure-changing sites. It is interesting to find out that DHX36 binding causes the largest structural decrease within 3′UTRs followed by CDSs and 5′UTRs (Fig. 2G), and the structure remodeling effect was conserved in both human HEK293T and mouse C2C12 cells (Fig. 2 and Fig. S6). More intriguingly, DHX36 binding-induced structure remodeling was not confined to localized binding sites (Fig. 2J) but extended across the entire transcript, especially in 3′UTRs (Fig. 2K, L). This phenomenon is reminiscent of the structure remodeling effect of some other RBPs, such as PUM2[58] and Dbp1[59], whose modulation also triggers structural changes on both RBP-bound and unbound sites, especially those in 3′UTRs. This also highlights the need to consider both localized changes and the entire transcript level when examining RNA structure

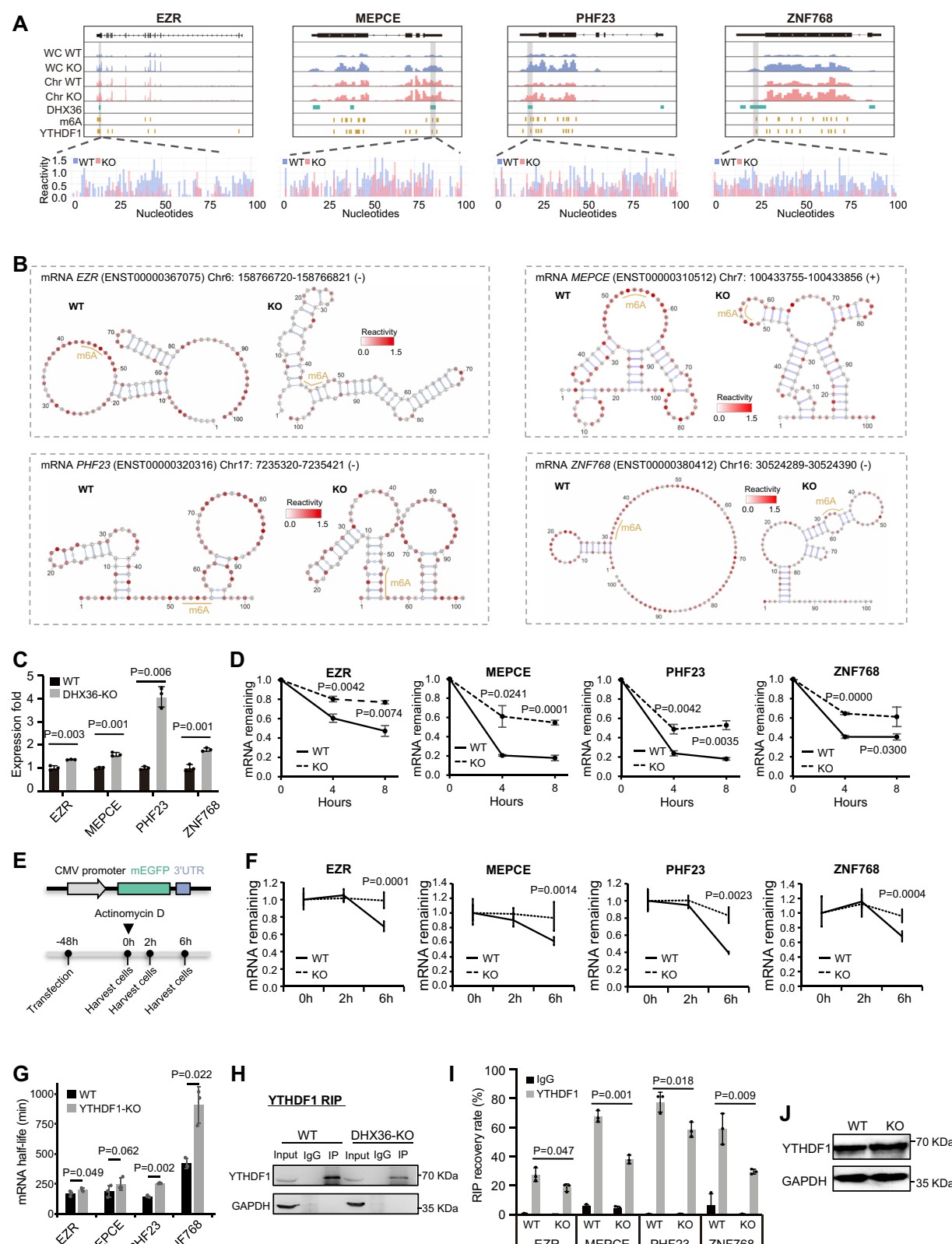

contribution to post-transcriptional regulation. Of note, our finding indicates that the dominant structural remolding effect in 3′UTRs is possibly caused by the low GC content (Fig. 2I). Low GC high AU content in the 3′UTRs causes low thermostability thus increased susceptibility to the unwinding process. rG4 binding by DHX36 is an ATP-independent process, but unwinding is energy-consuming, which involves a partial unfolding-refolding cycle[60]. Therefore, despite the

higher number of G4s in 5′UTRs compared to 3′UTRs and dense DHX36 binding (Fig. S2A, B), the 5′UTR G4s possess higher stability (Fig. S2C) and thus may be more resistant to DHX36 unwinding[61].

Our study also uncovers a previously uncharacterized mechanism through which DHX36 regulates mRNA abundance. We demonstrated that DHX36 binding-induced structural accessibility of 3′UTRs is involved in post-transcriptional regulation of RNA abundance (Fig. 4I).

**Fig. 6 | DHX36 facilitates YTHDF1 binding to decrease mRNA stability on selected targets. A** Genome tracks showing RNA abundance in chromatin fraction and whole cell, DHX36 binding sites, m⁶A sites, YTHDF1 binding sites, and SHAPE reactivity of the DHX36 binding sites in mRNAs EZR, MEPCE, PHF23, and ZNF768. **B** The folded structures of the DHX36 binding sites within the above mRNAs in WT and DHX36-KO cells. m⁶A and YTHDF1 binding sites are highlighted using yellow lines. Nucleotides are color-coded based on the SHAPE reactivity scores. **C** The abundances of the above mRNAs in WT and DHX36-KO cells were quantified by RT-qPCR ($n = 3$ biological replicates). **D** mRNA stability of the above mRNAs was determined by quantifying the mRNA abundance 0, 4, and 8 h after actinomycin D treatment ($n = 3$ biological replicates). **E** Schematic illustration showing the construction of the EGFP reporters. **F** The stability of the EGFP mRNAs fused with

DHX36 3'UTR binding site of each selected mRNA was determined by quantifying the EGFP mRNA abundance 0, 2, and 6 h after actinomycin D treatment ($n = 3$ biological replicates). **G** The half-life of the above mRNAs in WT and YTHDF1-KO. **H** RIP assay was performed in WT and DHX36-KO cells with a YTHDF1 antibody. The enrichment of the immunoprecipitated YTHDF1 and GAPDH proteins was assessed by Western blotting. IgG was used as a negative control. **I** The enrichment of the representative mRNAs in the above RIP was detected by RT-qPCR. The relative enrichment was normalized by input ($n = 3$ biological replicates). **J** Western blot confirmed the unaltered protein levels of YTHDF1 in WT and DHX36-KO cells. Data are presented as mean values ± SD in (**C**, **D**, **F**, **G**, **I**). The statistical significances in (**C**, **D**, **F**, **G**, **I**) were calculated by a two-sided Student's *t*-test. Source data are provided as a Source Data file.

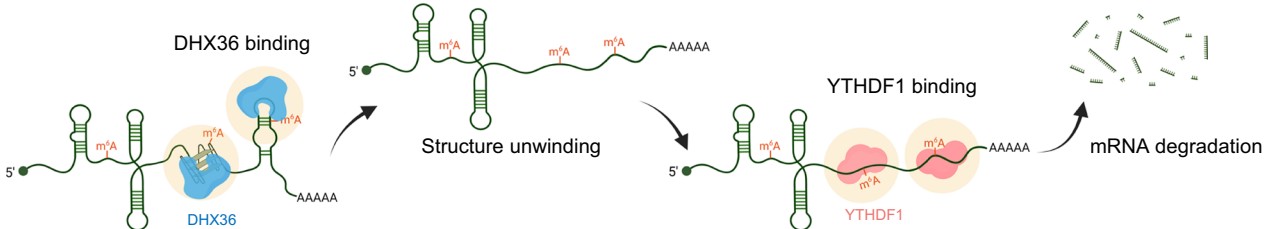

**Fig. 7 | Schematic illustration of the functional mechanism of DHX36-induced structure change in regulating the activity of YTHDF1 binding and target mRNA degradation.** DHX36 binds to and resolves the structured sites. The increased structural accessibility of DHX36-bound m⁶A sites promotes m⁶A-dependent YTHDF1 binding to promote target mRNA degradation. Created in BioRender. Zhang, Y. (2023) BioRender.com/u61b678.

The co-localization of DHX36 binding sites with m⁶A modification and YTHDF1 binding (Fig. 5B–D) especially around stop codons (Fig. 2C) inspired our mechanistic investigation of the role of m⁶A/YTHDF1 in mediating DHX36 effect on RNA degradation. Of note, Youn et.al detected subcellular co-localization of DHX36 and YTHDF1 in stress granules using in vivo proximity-dependent biotinylation analysis in HEK293 cell[62], further supporting the potential synergistic action between these two RBPs. DHX36 binding appears to exert limited influence on m⁶A deposition (Fig. 5H, I), suggesting that the inhibited YTHDF1 activity upon DHX36 loss is not caused by reduced m⁶A deposition. The known preference of the YTH domain towards less structured RNAs[33] led us to believe that DHX36 induced structural accessibility facilitates YTHDF1 recruitment to m⁶A sites. This was supported by our YTHDF1 CLIP-seq assays (Fig. 5J, K) and further validated by additional assays conducted on selected targets including EZR, MEPCE, PHF23, and ZNF768 (Fig. 6). Therefore, our findings suggest a previously unknown mechanism that DHX36 binding-induced RNA structural loss provides more accessible "landing pad" for YTHDF1, consequently promoting RNA degradation and reducing mRNA abundance (Fig. 7). It is also worth noting that in addition to YTHDF1, the binding of other factors could also be facilitated upon the increased structural accessibility to facilitate DHX36 regulation of post-transcriptional processes. For example, hnRNPC, which requires less structured m⁶A substrates[13] could also be recruited to function in modulating alternative splicing. It will thus be necessary to define structure-dependent DHX36 interactome in cells in the future. Altogether our findings support the emerging theme that RNA structure constitutes an important layer of gene regulatory determinants.

## Methods
### Cell culture
Human HEK293T cells (CRL-3216) and mouse C2C12 myoblast cells (CRL-1772) were obtained from American Type Culture Collection (ATCC) and cultured in DMEM medium (12800-017, Gibco) with 10% fetal bovine serum (FBS, 10270-106, Gibco) and 1% penicillin/streptomycin (P/S, 15140-122, Gibco) at 37 °C in 5% $CO_2$ incubator.

### Generation of DHX36 knockout cells
For human HEK293T cells, DHX36 gene exons were input into http://crispor.tefor.net, and crRNAs were generated and selected. crRNAs were ligated to the PX458 plasmid and synthesized by Genewiz (China). 0.1 million authenticated HEK293T cells were seeded on 24-well plates to culture in DMEM medium with 10% heat-inactivated fetal bovine serum, 100 units/ml of antibiotic anti-mycotic (Thermo Fisher Scientific, USA) at 37 °C in 5% $CO_2$ incubator. After 24 h, two 500 ng ligated PX458 plasmids were transfected into cells using Lipofectamine® 2000 (Thermo Fisher Scientific, 11668019) according to the manufacturer's protocol. 48 h later, cells were extracted and FACS-sorted (Sony SH800S cell sorter) according to the GFP signal, and positive cells were transferred to a new 96-well plate. Single clones were selected under a microscope and expanded in a six-well plate. After 4–7 days of incubation, cells were collected to extract DNA for genotyping and protein for knockout detection. Mouse DHX36 knockout C2C12 myoblast cells were generated and used as previously described[22,63].

### In vivo chemical probing
For HEK293T cells, the WT and DHX36-KO cells were harvested in 1X PBS buffer (Thermo Fisher Scientific, 10010049) and probed with NAI (50 mM in final) or 5% DMSO as negative control for 15 min at 37 °C, 500 rpm. Then DTT (200 mM in final) was added immediately to quench each probing reaction. After spinning down the cells (~500 g) for 3 min at room temperature, total RNAs were extracted and quantified using RNeasy Plus Mini Kit (Qiagen, 74136) and Nanodrop ND-1000 Spectrophotometer. For C2C12 cells, the proliferating WT or Dhx36-KO myoblasts were harvested in 1X PBS buffer and probed with DMS (20 mM in final) or 5% DMSO as negative control for 15 min at 37 °C. Then DTT (200 mM in final) was added immediately to quench each probing reaction. After spinning down the cells (~500 g) for 3 min at room temperature, total RNAs were extracted using RNeasy Plus Mini Kit (Qiagen, 74136).

## Gel-based identification of in vivo chemical probing for 5.8S rRNA

About 2 µg in vivo NAI-probed total RNA together with DMSO-treated total RNA were dissolved in 5.5 µl water, 1 µl of 5 µM Cy5-modified RT primer for 5.8S rRNA and 3 µl of reverse transcription buffer with dNTPs (20 mM Tris-HCl (pH 7.5), 4 mM MgCl₂, 1 mM DTT, 0.5 mM dNTPs, 150 mM LiCl in final) were denatured at 75 °C for 3 min. The reaction was cooled down to 37 °C for 5 min, and 0.5 µl SuperScript III (Thermo Fisher Scientific, 18080093) was added and mixed quickly. The RT reaction was incubated at 50 °C for 15 min following 0.5 µl of 2 M NaOH addition at 95 °C for 10 min to degrade the RNA template. 10 µl 2X denaturing dye (95% formaldehyde, 20 mM EDTA (pH 8.0), 20 mM Tris-HCl (pH 7.5), orange G) was added and incubated at 95 °C for 3 min. The resultant reaction was kept on 60 °C and loaded to 8% polyacrylamide-urea denaturing gel (containing 8.3 M of urea) for electrophoresis. For the sequencing lanes, RNAs were dissolved in 4.5 µl water; 1 µl 10 mM of ddATP/ddTTP/ddCTP/ddGTP (Roche, 12352200) was added at the beginning.

## Structure-seq library preparation

Detailed procedures for cDNA library preparation were described in two references[64] (for C2C12 libraries) and[65] (for HEK293T libraries) with slight modification. Briefly, each library was prepared with 500 ng poly(A)-selected RNAs after two rounds of selection (Thermo Fisher Scientific, AM1922). The poly(A) RNAs were fragmented to around 300-nt length at 95 °C for 1 min and purified using RNA clean and concentrator (Zymo Research, R1016). The fragmented RNA was then subjected to a 5′ dephosphorylation reaction at 37 °C for 30 min by adding rSAP (NEB, M0371L) and PNK enzyme (NEB, M0201S) in 1× T4 PNK buffer (NEB, B0201S). Next, the 3′adapter was ligated to the dephosphorylated RNA using T4 RNA ligase 2 KQ (NEB, M0373L) at 25 °C for 1 h. Excess adapters were digested and removed by adding RecJf (NEB, M0331) and 5′ deadenylase (NEB, M0331) for 30 min at 30 °C followed by column purification (Zymo research, R1016). The following reverse transcription (RT) was performed in 1X RT buffer (20 mM Tris, pH 7.5, 4 mM MgCl2, 1 mM DTT, 0.5 mM dNTPs, 150 mM LiCl) with SuperScript III enzyme (Thermo Scientific, 18080093) at 42 °C for 40 min. RNA was digested by 2 M NaOH, and the reaction was neutralized with 1 M Tris-HCl (pH 7.5) followed by column purification (Zymo Research, R1016). The 5′adapter (for C2C12 libraries) or dU adapter (for HEK293T libraries) was ligated to the cDNA with Quick T4 DNA ligase (NEB, M2200L) and incubated at 37 °C overnight. For the C2C12 library, the ligated product was run on the 10% denaturing urea-TBE acrylamide gel (Thermo Scientific, EC68752BOX), and the 100−400nt region was cut and dissolved in 1X TEN250 buffer (1X TE 7.4, 0.25 M NaCl) at 80 °C for 30 min followed by column purification (Zymo Research, R1016). For the HEK293T libraries, the ligated product was first purified by column (Zymo Research, R1016), then USER II enzyme (NEB, M5509) was added to cleave the product, followed by a second column purification (Zymo Research, R1016). The purified product was used for PCR cycle test and amplification with different barcoded reverse primers. PCR products with 150−400 bp were then cut and purified with 2% agarose gel and column (Zymo Research, D4008). Finally, all libraries were sequenced on the Illumina HiSeq System in 150 bp paired-end (PE) configuration.

## YTHDF1 CLIP-seq in WT and DHX36-KO HEK293T cells

CLIP-seq was performed as described previously with minor modifications[66–69]. Wild-type control (WT) or DHX36 knockout (KO) HEK293T ($2 \times 10^7$) were crosslinked by irradiation twice at 254 nm for 400 mJ/cm² in SCIENTZ03-II crosslinker (Ningbo Scientz Biotechnology). Cells were lysed in 1 ml Lysis Buffer [100 mM KCl, 5 mM MgCl2, 10 mM HEPES-KOH pH 7.0, 0.5% NP-40 supplemented with 1 mM DTT and protease inhibitor (A32965, Thermo Fisher Scientific)] for 15 min on ice with pipetting up and down for several times. Then, 5 µl of Turbo

DNase (AM2238, Thermo Fisher Scientific) and 10 µl of diluted RNase I (4 U µl⁻¹, 1:25 diluted in DPBS, AM2294, Thermo Fisher Scientific) were added to the reaction and incubated at 37 °C for 7 min with rotating at 1200 rpm. The cell lysate was immediately supplemented by 5.5 µl of Murine RNase inhibitor (R301-01, Vazyme) and cleared by centrifugation at 15,000 × g for 20 min. Then, 50 µl washed Dynabeads Protein G (10004D, Thermo Fisher Scientific) was added to each reaction to preclear the lysate and incubated for 30 min at 4 °C with rotation. About 20 µl of precleared samples were saved as input samples, and the remaining lysates were divided into two aliquots. To prepare IP buffer, 4,225 µl of ice-cold NT2 buffer (50 mM Tris-Cl pH 7.4, 150 mM NaCl, 1 mM MgCl₂, 0.05% NP-40) was supplemented with 1000 U Murine RNase inhibitor, 50 µl of 100 mM DTT and EDTA to 20 mM and 500 µl of precleared cell lysates were then added to the IP buffer. Immunoprecipitation was then performed using 10 µg anti-YTHDF1 antibody (17479-1-AP, Proteintech), or normal IgG control (2729, Cell Signaling Technology) respectively overnight at 4 °C with rotation. On the next day, 125 µl washed Dynabeads Protein G was added to each reaction and incubated for another 4 h with rotation. Beads were collected and washed twice with high salt wash buffer (50 mM Tris-Cl pH 7.4, 300 mM NaCl, 1 mM MgCl₂, 0.05% NP-40) and twice with wash buffer (20 mM Tris-HCl pH 7.4, 10 mM MgCl₂, 0.2% Tween-20). After the washing steps, 1/10 volume of beads were saved to determine immunoprecipitation efficiency. The remaining crosslinked RNA complexes were dephosphorylated by incubating with Quick CIP (M0525L, NEB) for 20 min at 37 °C, followed by treatment with T4 PNK enzyme (M0201L, NEB) in acid PNK buffer (70 mM Tris-HCl pH 6.5, 10 mM MgCl₂) for 20 min at 37 °C. Protein-RNA complexes were then divided into two portions. 1/10 volume was subjected to RNA 3′ end biotinylation following the procedure described previously[69]. The remaining was used for on-bead 3′ RNA linker ligation and incubated in a 25 µl reaction volume containing 120 pmol of 3′ RNA adapter (5′p-rGrArUrCrGrUrCrGrGrArCrUrGrUrArGrArArCrUrCrUrGrArArC-/3′ InvdT/) (Genewiz), 30 nmol of ATP, 25 U of T4 RNA ligase I (M0204L, NEB), 16 U Murine RNase inhibitor, 15% PEG 8000 and 3.6% DMSO at 16 ℃ for overnight, shaking at 1200 rpm. The beads (biotin-labeled or 3′ RNA adapter-ligated) were then denatured in loading buffer and resolved on a Novex NuPAGE 4–12% Bis-Tris gel and then transferred to 0.45 µm nitrocellulose membranes for 3 h at 30 V. The biotin reaction group was then subjected to biotin detection to indicate labeled RNA complexes. For CLIP-seq, labeled YTHDF1-RNA bands were excised from the membrane (starting from 65 kD to -65KD above the MW of YTHDF1) and eluted by incubating with PK buffer (100 mM Tris-Cl pH 7.4, 50 mM NaCl, 10 mM EDTA) and Proteinase K at 4 mg ml⁻¹ (P8107S, NEB) for 30 min at 37 °C, then for another 30 min at 37 °C with PK buffer, 3.5 M Urea and Proteinase K. RNA was subsequently purified using acidic phenol:chloroform (P1023, Solarbio) and ethanol precipitation. Then the RNA was 5′ ends phosphorylated with T4 PNK and then ligated to 5′ RNA adapter (5′-rCrCrUrUrGrGrCrArCrCrCrGrArGrArArUrUrCrCrA-3′) (Genewiz) in a 10 µl reaction volume containing 50 pmol of 5′ RNA adapter, 10 nmol of ATP, 10 U of T4 RNA ligase I (M0204L, NEB), 20 U Murine RNase inhibitor and 10% PEG 8000 at 16 °C for overnight. Following the ligation, ligated RNA was purified by the acidic phenol:chloroform (P1023, Solarbio) and ethanol precipitation. The resultant RNA was reverse transcribed with RP1 primer (5′-AATGATACGGCGACCACCGAGATCTACACGTTCAGAGTTCTACAG TCCGA-3′) using HiScript IV 1st Strand cDNA Synthesis Kit (R412-01, Vazyme). A small portion of cDNA was serially diluted and subjected to test PCR amplification to determine the optical PCR cycle. The full-scale PCR amplification was performed using 2 × Phanta Max Master Mix (P515-01, Vazyme) with 12.5 pmol of RP1 primer and RPI-index primers (5′-CAAGCAGAAGACGGCATACGAGATNNNNNNGTGACTGGA GTTCCTTGGCACCCGAGAATTCCA-3′, in which "NNNNNN" is an index sequence). The optimal PCR cycle for full-scale amplification is 17. PCR products were PAGE separated, and DNA from 140 bp to 350 bp was

eluted from PAGE gel. The library was quantified and sequenced on the Illumina NovaSeq X Plus (Shanghai Sequanta Technologies).

## RNA bind-n-seq (RBNS) assay

RHAU53 peptide with a His tag at the C terminal was synthesized by Synpeptide Co Ltd and stored at −80 °C. To perform the RBNS assay, we followed a modified version of the method described previously[70,71] with modification. First, we generated a dsDNA template from a single-stranded 40 nt random DNA and reverse extension primer through an extension reaction using Superscript III reverse transcriptase (Thermo Fisher Scientific, 18080085). RNAs with a random region of 40 nt were prepared by in vitro transcription (NEB, E2040S) using the dsDNA as a template. We then heated the concentrated RNAs at 85 °C for 5 min in the presence of 150 mM KCl or LiCl and allowed them to cool for at least 3 min to promote structural folding. Then the RNA library was incubated with an increasing concentration of RHAU53 (5, 20, 80, 320, and 1300 nM) with binding reactions as previously described[71], with slight modifications. In this study, 10 μl NEBExpress ® Ni-NTA Magnetic Bead (NEB, S1423L) was used to replace the Dynabeads MyOne Streptavidin T1 in the old method to pull down the His-tagged protein-RNA complexes. All wash and binding buffers contained either 150 mM KCl or LiCl. Protein-RNA complexes were eluted in 1X elution buffer (20 mM Tris-HCl (pH 7.5), 150 mM KCl or LiCl, 500 mM Imidazole (pH 7.4), 8 M Urea). RNAs were isolated as previously described[71] and reverse transcribed for PCR amplification and sequencing (See Structure-seq library preparation Method).

## mRNA stability assay

WT or DHX36-KO HEK293T cells were treated with 5 μg/ml actinomycin D (Thermo Fisher Scientific, 11805017) for 0, 4, and 8 h separately. Cells were collected for total RNA isolation using the RNeasy Plus Mini Kit (Qiagen, 74136). The same amount of total RNAs from each sample were reversely transcribed with PrimeScript RT Master Mix (TaKaRa, RR036A). Diluted cDNAs were mixed with primers and SsoAdvanced Universal SYBR® Green Supermix (BioRad, 1725272) for qPCR amplification with CFX Connect Real-Time PCR System (BioRad). The GAPDH gene was used for normalization. For the stability reporter assay, the DHX36 binding region on the 3′UTR of the four target mRNAs were cloned to the p-EGFP-N1 expression plasmid at the downstream of the EGFP open reading frame, respectively. The reporter vectors were transfected to WT or DHX36-KO cells, and 48 h later, the cells were treated with 5 μg/ml of actinomycin D for 0, 2, or 6 h separately. Cells were then collected for total RNA extraction, and qRT-PCR was conducted to examine the level of EGFP mRNA with GAPDH mRNA used as the internal control.

## RNA immunoprecipitation assay

WT or DHX36-KO 293T cells were washed with 1X PBS two times and collected by centrifugation at 700 × g, 4 °C for 5 min. Cell pellets were lysed with RIP lysis buffer (10 mM HEPES, pH 7.0, 100 mM KCl, 5 mM MgCl₂, 0.5% NP-40, 1 mM DTT, 100 units RNase-out, and 1x protease inhibitor cocktail) on ice for 10 min. The supernatant was collected after centrifugation at 14,000×g at 4 °C for 15 min. In total, 5 μg of antibodies against YTHDF1 (Proteintech 17479-1-AP) or isotype IgG (Santa Cruz Biotechnology, sc-2025) were incubated with 30 μl of Protein G DynabeadsTM (Thermo Fisher Scientific, 10003D) with rotation at RT for 1 h followed by adding same volume of cell lysate to the lysate with further incubation with rotation at 4 °C overnight. The beads were washed with ice-cold RIP wash buffer (50 mM Tris-HCl pH 7.4, 150 mM NaCl, 1 mM MgCl₂, 0.05% NP-40) six times, of which 1/3 volume were removed from the bead suspension during the last wash for western blotting. Proteinase K was then added to the beads with incubation at 4 °C for an hour. After discarding the supernatant from the final wash, Qiazol lysis reagent was added (Qiagen) to the sample

for RNA extraction. Extracted RNAs were resuspended in 30 μl of RNase-free water, and cDNAs were obtained from reverse transcription for the qPCR test (See above). 10% of cell lysates of each sample was collected, followed by total RNA extraction and qPCR test for normalization.

## m⁶A inhibition and meRIP-qPCR

HEK293T cells were treated with 40 μM METTL3 inhibitor STM2457 (MCE, HY-134836) for 24 h. The decreased m⁶A modification was validated by meRIP-qPCR assay as previously described[72]. In brief, 2.5 μg RNA was diluted in 250 μl IP buffer (10 mM Tris-HCl, pH 7.4, 150 mM NaCl, 0.1% NP-40, and Protease Inhibitor Cocktail [Sigma-Aldrich]). About 50 μl of the diluted RNA was used as 25% input. About 20 μl Dynabeads protein G (Thermo Fisher/Life Technologies) was washed three times with 1 ml IP buffer and incubated with 2 μg m⁶A antibody (Abcam, ab151230) at RT for 30 min. The conjugated Dynabeads-m⁶A-antibody was washed in 1 ml IP buffer three times and resuspended in 800 μl IP buffer. Then 200 μl diluted RNA sample was mixed with antibody-protein G beads and rotated at RT for 3 hr. After 3 h rotation, the beads were washed with IP buffer five times and resuspended in 50 μl IP buffer. Both immunoprecipitated and 25% input RNAs were extracted from the beads by TRIzol. The extracted RNA was subjected to cDNA synthesis (Vazyme, HiScript IV RT SuperMix for qPCR) and qPCR (Roche, lightcycler 480 Instrument II) to quantify the m6A levels.

## Western blotting (WB)

Cells were harvested and lysed in RIPA reagent (Thermo Fisher Scientific, 89900) for 15 min. on ice. Total proteins were centrifuged (18,000 × g at 4 °C) and determined using protein assay dye reagent (Bio-Rad, 500006). Subsequently, 25 μg proteins were separated via 4–10% SDS-PAGE and transferred to PVDF membranes. Following blocking with 5% non-fat milk for 1 h at room temperature, the membrane was incubated overnight at 4 °C with primary antibody dilution. After washing three times with 1X TBS-T buffer (150 mM NaCl, 20 mM Tris-HCl (pH 8.0), and 0.05% Tween-20) for 10 min, the membrane was incubated with secondary antibody dilution for 1 h at room temperature. Subsequently, the membrane was washed three times with 1X TBS-T buffer for 10 min. After staining with Clarity Max Western ECL Substrate (Bio-Rad, 1705062), the signal was visualized using Chemi-Doc™ Touch Imaging System. The primary antibodies used are as follows: DHX36 (Proteintech 13159-1-AP) and GAPDH (Santa Cruz, sc-32233). Secondary antibodies used are Goat anti-Mouse IgG (H/L): HRP (Bio-Rad, STAR207P) and Goat anti-Mouse IgG (H/L): HRP (Beyotime, A0208).

## Structure-seq data analysis

Both structure-seq data in HEK293T and C2C12 cells were processed using StructureFold2[35]. The adapters were first removed using Cutadapt v1.15[73], and the reads shorter than 20nt or with a quality score lower than 30 were filtered out. The remaining reads were mapped to human or mouse transcriptome reference using Bowtie2 v2.3.3.1[74] with default parameters. Human and mouse transcriptome references were respectively obtained from GENCODE v33 and vm24 database[75]. Next, the mapped reads with more than three mismatches or mismatches at the first base were removed using the sam_filter.py script. For each library, RT stops were counted using the sam_to_rtsc.py script. The correlation of RT stops between two replicates was determined using the rtsc_correlation.py script after removing the outliers (RT stops >10e5). The average RT coverage of each transcript in NAI or DMS-treated libraries was calculated using the rtsc_coverage.py script. For NAI-treated libraries, average coverage was calculated as the sum of RT stops within the transcript divided by the length of the transcript. For DMS-treated libraries, only adenines and cytosines were

considered. Transcripts with average coverage higher than 1 in all replicates of all the treated WT and KO samples were saved, followed by the combination of RT stop counts across two biological replicates from each sample. The raw reactivity score for each nucleotide was calculated based on the modified formula as Yang et al. described[76]:

$$Reactivity_i = \frac{\ln(1+P_i)}{\sum_i \ln(1+P_i)/L} - \alpha \frac{\ln(1+M_i)}{\sum_i \ln(1+M_i)/L} \quad (1)$$

$$\alpha = \min(1, \sum_i \ln(1+P_i)/\sum_i \ln(1+M_i)) \quad (2)$$

where $P_i$ and $M_i$ respectively represent the RT stops of nucleotide $i$ in NAI/DMS-treated and DMSO-treated samples; $L$ represents transcript length; $\alpha$ is a library size correction factor. The raw reactivity was calculated using log-normalized RT stops and then normalized by 2–8% method to generate the final reactivity. In the normalization process, the top 10% of raw reactivities were extracted. Within this subset, the top 2% of raw reactivities were excluded, and the remaining 8% of reactivities were averaged to generate a normalization scale[77]. Subsequently, the raw reactivities of both WT and KO samples were divided by the WT normalization scale to generate normalized reactivities. Next, the splice_reacts_by_FASTA.py script was used to splice the reactivity profile of the entire transcript by the designated mRNA regions. The average reactivity and the Gini index of SHAPE reactivity of the entire transcript or the designated region were calculated using the react_statistics.py script. RNA structure folding constrained by reactivity score was performed using the Vienna RNA package[41]. The base-pairing probability of each nucleotide was extracted from the corresponding structure ensemble ".ps" file. The software dStruct[40] (parameter: reps_A = 2, reps_B = 2, min_length = 5, batches=T, check_signal_strength = T, check_nucs = T, check_quality = T) was used to identify the significant DRRs within DHX36 binding sites (FDR < 0.25). VARNA[78] was used to visualize the selected RNA secondary structures (Figs. 3H, 6B).

For 18S rRNA benchmarking, we respectively aligned the reads in HEK293T or C2C12 cells to human and mouse 18S rRNA reference (human: NR_145820.1; mouse: NR_003278.3). The RT stops and reactivity scores were calculated based on the same workflow as stated above. Finally, the reactivity scores were mapped to the phylogenetic structures of human or mouse 18S rRNA, which were obtained from CRW database[79], to evaluate the performance of our in vivo structure-seq libraries.

**Analysis of PAR-CLIP and CLIP-seq data for identification of DHX36 binding sites**
Publicly available PAR-CLIP data (GSE105171)[37] were analyzed to identify the endogenous DHX36 binding sites. Since the use of 4-thiouridine (4SU) labeling on the transcripts restricts crosslinking to a single base and leads to T-to-C mutation during reverse transcription of crosslinked RNAs, the pipeline described in the CLIP Tool Kit (CTK)[38] was used to identify crosslinking sites at single-nucleotide resolution. Briefly, 3'adapter sequences and low-quality reads were trimmed by Cutadapt v1.15[73] (parameter: -n 1 -q 5 -m 20), followed by collapsing the exact duplicates with the same sequence. Then, we mapped the remaining reads to human hg38 genome reference using BWA v0.7.17[80] (parameter: -n 0.06 -q 20). To achieve peak calling at single-nucleotide resolution, the crosslinking-induced mutation sites (CIMS) mode of CTK was used. To improve the reliability, we defined the significant CIMS with FDR <0.05 in both biological replicates as exact crosslink sites and the ±50 nt of crosslink sites as DHX36 binding sites. DHX36 binding sites were annotated to the 5'UTR, CDS, or 3'UTR regions based on the GENCODE human v33 annotation file[75].

CLIP-seq data (GSE151124) in C2C12 cells were analyzed using the same pipeline as previously described[22]. The adapters and low-quality reads were trimmed using FastQC[81] and Trimmomatic[82]. Following duplicate removal, the remaining reads were aligned to mouse genome mm10 using Bowtie2 v2.3.3.1[74], and peaks were called using Piranha v1.2.1[83] (parameter: -p_threshold 0.05 -b 100). Annotation of the peaks to designated mRNA regions was based on GENCODE mouse vm24 annotation file[75].

**RNA bind-n-seq data processing**
RBNS data were processed using RBNS pipeline[71]. Briefly, 6mer motif frequency was calculated in each RHAU53-bound pool at 5 different concentrations (5, 20, 80, 320, and 1300 nM) and the input pool. The motif R value representing the enrichment level was defined as the frequency of each 6mer in the RHAU53-bound pool divided by the frequency in the input pool.

**rG4 prediction and evaluation**
rG4 prediction and classification were performed using our published strategy[22]. The script can be accessed in the GitHub repository (https://github.com/jieyuanCUHK/DHX36_paper). rG4 score for each rG4 was evaluated by the R package pqsfinder[84] (parameter: strand = "+", run_max_len = 3 L, loop_min_len = 1 L, loop_max_len = 21 L, min_score = 1 L). In this scoring system, rG4s with 3 G-quartets, shorter loops, and fewer bulges were assigned with higher scores because of their high stability[85].

**PARIS data analysis for identification of long-range interactions**
PARIS data in HEK293T cells (GSE74353) was obtained and analyzed following the published study[39]. Briefly, the adapters of raw reads were trimmed using Trimmomatic[82], followed by the removal of PCR duplicates. The remaining reads were mapped to the human hg38 reference using STAR[86]. Next, the secondary mappings were removed using Samtools[87]. Finally, the arms (two RNA fragments of each PARIS read) of the long-range RNA base pairing were identified using the publicly available scripts (https://github.com/qczhang/paris).

**RNA-seq data analysis**
RNA-seq data from the chromatin fraction or whole-cell in WT and DHX36-KO HEK293 cells were used in our study (GSE105175)[37]. All these data were analyzed using the same pipeline. First, the 3' adapters and low-quality bases were trimmed by Cutadapt[73]. Next, the remaining reads were mapped to human hg38 genome reference using HISAT2 v2.0.4[88] with default parameters. The uniquely mapped reads were then counted using featureCounts v2.0.1[89]. To obtain the DHX36 loss-induced differential expression profile, we used the R package DESeq2[90] to normalize the read counts of each gene and evaluate the statistical significance and the fold change of the RNA abundance upon DHX36 loss in both whole-cell and chromatin fractions. The post-transcriptional fold change (pFC) was defined as the fold change in chromatin fraction divided by the fold change in the whole cell. The genes with pFC greater than 1.5 or lower than 1/1.5 were defined as post-transcriptionally up- or down-regulated genes.

**Analysis of miCLIP data for identification of m⁶A modification**
Publicly available miCLIP data in WT HEK293 (GSE63753)[48] and HEK293T cells (GSE122948)[91] were used to identify m⁶A modification at single-nucleotide resolution. For miCLIP in HEK293 cells, both Synaptic Systems (SySy) and Abcam libraries were collected and analyzed as demonstrated[48]. First, the raw reads were trimmed using Cutadapt v1.15[73] and demultiplexed based on the experimental barcodes using pyCRAC tool[92]. Next, the replicates were combined, followed by PCR duplicate removal. For the paired-end data of Abcam libraries, reverse reads were reverse complemented and assigned with random

barcodes. The processed reads were mapped to human hg38 genome reference using BWA v0.7.17[80]. The CIMS pipeline in CTK[38] was used to identify the C to T mutations. For single-end data of SySy libraries, following read mapping, the CITS pipeline in CTK[38] was used to call truncation sites. The significant C to T mutation and truncation sites were selected as the exact crosslink sites. The crosslink sites identified in either Abcam or SySy libraries were defined as the m⁶A-modified sites. For miCLIP data in HEK293T cells, the same analytical workflow for Abcam libraries was used to identify significant crosslink sites. The crosslink sites identified by at least one miCLIP data were defined as m⁶A sites.

### Analysis of YTHDF1 iCLIP data and CLIP-seq data

Identification of YTHDF1 binding sites was based on iCLIP data in HEK293T cells (GSE78030). First, Cutadapt v1.15 was used to trim the raw reads and filter out the low-quality reads. Next, the exact duplicates with the same sequences were removed, and the remaining reads were mapped to the human hg38 genome reference using BWA v0.7.17[80]. The uniquely mapped reads were saved, and potential PCR duplicates were removed to obtain unique CLIP tags. The unique tags for each replicate were merged, and the CITS pipeline in CTK was used to identify crosslink-induced truncation sites. The significant truncation sites were defined as YTHDF1 binding sites ($P < 0.05$). YTHDF1 CLIP-seq data generated in WT and DHX36-KO HEK293T cells were analyzed using a similar pipeline, including adapter trimming using Cutadapt, mapping using BWA, duplicate removal, peak calling using the CIMS pipeline in CTK. Next, deepTools[93] was used to compare the signals of YTHDF1 binding within DHX36 binding sites between WT and DHX36-KO samples.

### Motif analysis of DRR regions, m⁶A-modified sites, and YTHDF1 binding sites

The sequence motifs of the DHX36-induced DRR regions were discovered by HOMER[94] (parameter: -size 5 -len 5,6,7 -norevopp -rna). Next, the structure of each nucleotide was classified into "paired" (denoted as P) and "unpaired" (denoted as U) according to the dot-bracket notation generated by Vienna RNA package partition function with SHAPE reactivities as constraints, and MEME[95] in the differential enrichment mode was used to discover the differential structural motifs of the DRRs between WT and DHX36-KO samples. Single-nucleotide m⁶A-modified sites and YTHDF1 binding sites were extended for 5nt for both sides. Next, HOMER[94] was used to discover the m⁶A and YTHDF1 motifs de novo within the extended sites (parameter: -size 10 -len 5 -norevopp -rna) and determine the enrichment of m⁶A and YTHDF1 motifs within 3′UTR binding of DHX36 (parameter: -len 5,6,7,8,9,10 -mknown <m⁶A/YTHDF1 motif> -norevopp -rna).

### Analysis of 4SU-TT-seq data for identification of YTHDF1 degradation targets

The half-lives of mRNAs in WT and YTHDF1-KO HEK293 cells were obtained from the published 4SU-TT-seq data (GSE143994)[25]. The fold changes of half-lives (YTHDF1-KO/WT) in three biological replicates were respectively calculated and then averaged to overall fold changes. Finally, we defined the mRNAs with overall fold changes greater than 1 as YTHDF1-induced degradation mRNAs.

### Functional enrichment analysis

R package clusterProfiler[96] was used to perform the gene ontology (GO) analysis on the mRNAs that are most strongly affected by the DHX36-induced structural changes. All default settings and FDR corrections were used.

### Oligonucleotides

The oligonucleotides used in this study can be accessed in Supplementary Data 9.

### Reporting summary

Further information on research design is available in the Nature Portfolio Reporting Summary linked to this article.

## Data availability

The data supporting the findings of this study are available from the corresponding authors upon request. Structure-seq and RBNS data used in this study have been deposited in Gene Expression Omnibus (GEO) database under the accession codes GSE237160 (WT and DHX36/Dhx36 Structure-seq), GSE237161 (RBNS), GSE264498 (YTHDF1 CLIP-seq), and GSE264642 (Control and m⁶A-inhibited Structure-seq). Source data for the figures and Supplementary Figs. are provided as a Source Data file. Source data are provided with this paper.

## Code availability

The code used in this study is available at Zenodo[97] and the GitHub repository (https://github.com/zhangyw0713/Scripts_for_DHX36_paper).

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

## Acknowledgements

This work was supported by National Key R&D Program of China to H.W. (2022YFA0806003); the National Natural Science Foundation of China (NSFC) (grant no. 82172436 to H.W.; 32300703 to X.C.; 32270587 to Y.Z.; 32471343 to C.K.K); National Natural Science Foundation of China (NSFC) Excellent Young Scientists Fund (Hong Kong and Macau) Project (32222089) to C.K.K.; Natural Science Foundation of Guangdong Province, China to X.C. (2024A1515030291); General Research Funds (GRF) from the Research Grants Council (RGC) of the Hong Kong Special Administrative Region (14115319, 14100620, 14106521, and 14105823 to H.W.; 14120420, 14103522, and 14105123 to H.S.; RFS2425-1S02, CityU 11100123, CityU 11100222, and CityU 11100421 to C.K.K.); the research funds from Health@InnoHK program launched by Innovation Technology Commission, the Government of the Hong Kong SAR, China to H.W.; Collaborative Research Fund (CRF) from RGC to H.W. (C6018-19GF); Theme-based Research Scheme (TRS) from RGC (project number: T13-602/21-N); Area of Excellence Scheme (AoE) from RGC (project number: AoE/M-402/20); Health and Medical Research Fund (HMRF) from Health Bureau of the Hong Kong Special Administrative Region, China (project Code: 10210906 and 08190626 to H.W.); Croucher Foundation Project (9509003) to C.K.K.; State Key Laboratory of Marine Pollution Seed Collaborative Research Fund (SCRF/0037, SCRF/0040, SCRF0070) to C.K.K.; City University of Hong Kong projects (7030001, 9678302 and 6000827) to C.K.K.; the Hong Kong Institute for Advanced Study, City University of Hong Kong [9360157] to C.K.K.

## Author contributions

Y.W.Z., X.C., and H.W. conceived the research. Y.W.Z. designed and performed all computational analyses. K.L. constructed the DHX36-KO cell line. J.Z. constructed Structure-seq library, RBNS library, and performed experimental validation assisted by X.C. Y.Q. conducted m6A inhibition and quantification. J.K. and Y.Z. constructed YTHDF1 CLIP-seq

libraries. F.Y. analyzed the YTHDF1 CLIP-seq data. X.G. conducted the reporter assay assisted by X.C. H.S. supervised computational analyses. Y.D. and C.K.K. supervised RNA Structure-seq and RBNS experiments and analyses. C.K.K. and H.W. supervised experimental validation. Y.W.Z., J.Z., X.C., C.K.K., and H.W. wrote the manuscript with input from all authors.

## Competing interests

The authors declare no competing interests.
