## [Transparent Peer Review file · Nature Communications]

DHX36 binding induces RNA structurome remodeling and regulates RNA abundance via m6A reader YTHDF1

Corresponding Author: Professor Huating Wang

Version 0:

Reviewer comments:

Reviewer #1

(Remarks to the Author)

The manuscript titled "DHX36 binding induces RNA structurome remodeling and regulates RNA abundance via m6A/YTHDF1" by Zhang et. al explores the role of the DEAH-box helicase 36 (DHX36) in modulating RNA structures and its impact on RNA fate. They find that DHX36 binding leads to increased structural accessibility in the 3'UTRs, which correlates with decreased post-transcriptional mRNA abundance. Further analyses and experiments reveal that DHX36 binding sites are enriched for N6-methyladenosine (m6A) modification and YTHDF1 binding. The authors propose that DHX36-induced structural changes may facilitate YTHDF1 binding to m6A sites, resulting in RNA degradation. Overall, the findings contribute to our understanding of the role of RNA structures in mediating RNA binding protein functions in post-transcriptional regulation. There are several points that need clarification and further investigation.

1. The authors should address whether any normalization procedures or internal RNA structure controls were employed to account for potential systematic differences across samples in the analysis involving cross-sample comparisons of structural probing profiles. This information is crucial for interpreting the results accurately.
2. It would be informative to investigate the structural changes in DHX36-unbound mRNAs and non-binding sites of bound mRNAs to understand the specificity and global impact of DHX36 binding on RNA structures.
3. The authors should explore whether the DHX36-binding sites exhibit enriched long-range RNA-RNA interactions with distal regions. This analysis could shed light on potential long-range structural rearrangements induced by DHX36 binding.
4. A comprehensive analysis of the sequence and/or structural motifs enriched in the differentially remodeled regions (DRRs) of DHX36 wild-type vs. knockout cells would provide insights into the consequences of DHX36 depletion and potential cellular and physiological effects.
5. There appears to be confusion regarding the terminology used to describe the loss of RNA structures. For instance, the title of the first subsection suggests a "global loss of RNA structures" induced by DHX36 depletion, but the text indicates that DHX36 depletion actually leads to structural increase. The authors should clarify and ensure consistency in the usage of terminology throughout the manuscript.
6. The correlations presented in many figures (e.g., Figure 4F-I, particularly G and H) appear relatively weak (here in Figure 4 indicating subtle effects of DHX36 knockout on most transcripts). It would be beneficial to perform additional analyses to identify subsets of transcripts that are more strongly affected by DHX36 loss.
7. I suggest investigating the translational changes upon DHX36 loss and exploring the role of RNA structures in this process. Given the involvement of YTHDF1 in translational regulation, it would be interesting to examine its participation in the context of DHX36-mediated effects.

Reviewer #2

(Remarks to the Author)

In this manuscript, Zhang and colleagues propose a new mechanism of regulation by DHX36, and its interplay with m6A. The story is potentially interesting, but it raises a number of questions. I have here summarized the main points:

- High reproducibility was observed: 0.75-0.82. I would like to stop perpetuating the claim made by several groups that correlations below 0.9 are "high correlations" when it comes to chemical probing, because that is definitely not the case. In some cases apparent lower correlations can be due to the fact that certain bases exhibit extremely high reactivities, therefore using a Spearman correlation or discarding outliers above a certain RT-stop intensity, might help revealing the actual correlation. Also, was the correlation calculated by including or excluding the lowly expressed transcripts? Those might negatively affect correlation, but they do not reflect the authentic reproducibility of the experiment.
- Agreement between rRNA structures and SHAPE data must be evaluated as a AUROC instead
- What statistical test was used in Fig. 1E? The medians do not look different, suggesting that only a tiny fraction of transcripts are actually affected by the loss of DHX36. Was the Gini calculated along the full transcripts? What would happen if breaking down the transcripts into deciles, or simply into 5'UTR, CDS and 3'UTR and performing the same analysis? Same for Fig. 2E.
- The binding of DHX36 to both 5'UTRs and 3'UTRs might be a consequence of the presence of long-range interactions in mRNAs, between these regions. Upon crosslinking the interacting portions would be both captured during CLIP. It is possible to speculate that DHX36 regulates the switch between long-range base-pairing and local G4 formation. In this regard, I would suggest the authors to analyze previously published PARIS in HEK293 cells (Chang's lab) and other RNA duplex capture data to see if these regions are indeed enriched for long-range interactions. The same can also be investigated in terms of base-pairing probabilities using the ViennaRNA package partition function, using either the SHAPE reactivities from the WT or KO cells as constraints, to see if a significant difference in the distribution of base-pair distances exists.
- Is m6A presence necessary for DHX36 binding to RNA? Does DHX36 sense directly the presence of the modification, or does the m6A deposition lead to a structural change that makes DHX36 binding possible? I would encourage the authors to evaluate this (either transcriptome-wide or on a subset of targets), by checking DH36 binding following METTL3/14 knockdown/knockout.

Reviewer #3

(Remarks to the Author)

In this manuscript, Zhang et al. investigate the alteration of the RNA structure induced by DHX36 depletion. DHX36 is shown to open RNA structures at 3'UTRs, which correlates with decreased RNA abundance. Furthermore, N6-methyladenosine (m6A) modification sites and YTHDF1 binding sites are found to be enriched in DHX36 binding regions. The structural changes in RNA induced by DHX36 may facilitate YTHDF1 binding to m6A sites, subsequently leading to RNA degradation. Overall, this study profiles the changes in the DHX36-induced RNA structure using NAI detection and demonstrates its impact on RNA abundance through the regulation of YTHDF1 binding. The work is well executed, and the results are presented clearly. However, I do have some questions and suggestions for the authors:

1. Fig. 1c demonstrates that NAI probing is unbiased in 5.8s rRNA, as shown by gel analysis. To validate the unbiased nature of RNA structure sequencing data in different treatment groups, the authors should exhibit conserved mRNA elements or rRNA structures in different treatment groups using sequencing data.
2. In Fig. 1d, the global average SHAPE scores of mRNA are significantly decreased (FC=0.74). In Fig. 2e, it is shown that the fold change (FC) of DHX36-bound mRNA is 0.68. It would be helpful to explain why the RNA structural changes in DHX36-bound mRNA are less pronounced than global changes.
3. Fig. 2i indicates that the significant structural changes in 3'UTRs are possibly caused by low GC content. However, DHX36 is known to unwind RNA rG4 structures. It would be interesting to explore the reasons behind these extensive structural changes in 3'UTRs, especially in the context of low GC content.
4. The author's findings suggest more localized structural effects surrounding DHX36 binding sites (Fig. 3c). To strengthen this observation, it would be beneficial to present longer regions that exhibit these effects, rather than just focusing on the +/- 50nt of the crosslink site.
5. It would be valuable for the authors to investigate whether DHX36 binds to mRNA JUN. If it does, the binding site should be labeled on Fig. 3h for clarity.
6. Regarding the canonical m6A and YTHDF1 binding motifs, they typically follow the RRACH motif, especially for conserved adenosine. It appears that the DHX36 3'UTR binding sites do not show enrichment for the m6A binding motif (Fig. 3b). The author should do YTHDF1 iCLIP in WT and DHX36 KO groups, and combine RNA structure data/m6A sites, to prove DHX36 regulate YTHDF1 binding by RNA structure switch in m6A-dependent manner.
7. Previous studies have found that m6a can unwind local structures. The author should perform an m6a deletion assay to demonstrate that the structural changes are induced by DHX36, rather than by m6a.
8. In Figure 6, the author should conduct reporter gene experiments to validate their findings and molecular mechanisms, rather than solely relying on qPCR to verify these sequencing data.

Version 1:

Reviewer comments:

Reviewer #1

(Remarks to the Author)

The revision has addressed some of my concerns, but there are still a few outstanding questions.

1.1 I am curious about the impact of the normalization approaches. Simply stating that a script was run is not enough detail. Could you provide more information on how the normalization was effective?

1.2 The authors presented interesting findings in their response. However, I am specifically interested in understanding the structural differences between the DHX36-unbound mRNAs and the non-binding sites/regions within the DHX36-bound mRNAs.

1.6 Is there any additional insight or findings from analyzing the subsets of mRNAs that are most strongly affected by the DHX36-induced structural changes? I'm curious to understand more about the functional roles and regulatory mechanisms associated with these transcripts.

1.7 I appreciate the effort to include the analysis of published polysome profiling data from WT and DHX36-KO C2C12 cells. However, since these are different cell lines, the results are not entirely convincing. I would suggest either removing these results or performing the polysome profiling analysis directly in the HEK293 cells, to provide more meaningful and relevant data.

Reviewer #2

(Remarks to the Author)

The authors addressed all my comments. I am satisfied with their answers and the additional experiments they performed.

Reviewer #3

(Remarks to the Author)

The authors have addressed most of my concerns. I recommend its publications.

Point to point response

The manuscript titled "DHX36 binding induces RNA structurome remodeling and regulates RNA abundance via m6A/YTHDF1" by Zhang et. al explores the role of the DEAH-box helicase 36 (DHX36) in modulating RNA structures and its impact on RNA fate. They find that DHX36 binding leads to increased structural accessibility in the 3'UTRs, which correlates with decreased post-transcriptional mRNA abundance. Further analyses and experiments reveal that DHX36 binding sites are enriched for N6-methyladenosine (m6A) modification and YTHDF1 binding. The authors propose that DHX36-induced structural changes may facilitate YTHDF1 binding to m6A sites, resulting in RNA degradation. Overall, the findings contribute to our understanding of the role of RNA structures in mediating RNA binding protein functions in post-transcriptional regulation. There are several points that need clarification and further investigation.

1.1. The authors should address whether any normalization procedures or internal RNA structure controls were employed to account for potential systematic differences across samples in the analysis involving cross-sample comparisons of structural probing profiles. This information is crucial for interpreting the results accurately.

Thanks for your comment. We agree that the normalization or controls are important for the analyses. In fact, in the initial submission, we did perform the normalization procedure for the Structure-seq data but forgot to include the information in Methods. It is now in the revised Methods on page 33. Briefly, the normalization was conducted using the <.scale> file generated by the rtsc_to_react.py script of StructureFold2 package.

1.2. It would be informative to investigate the structural changes in DHX36-unbound mRNAs and non-binding sites of bound mRNAs to understand the specificity and global impact of DHX36 binding on RNA structures.

Thanks for the critical comment. In our initial submission, the comparison of the reactivity or Gini index between DHX36-bound and unbound mRNAs were included (revised Fig. 2F, Text page 9). We have now also added the comparison on each mRNA region (revised Fig. S3B, Text page 9). Additionally, as suggested we have also compared the localized reactivity change between the DHX36 binding sites and the randomly selected sites on the same mRNA. As shown in the revised Fig. S4F-H (Text page 12), 5'UTR and CDS binding sites showed significantly higher structural changes compared to random sets, whereas DHX36 binding sites at 3'UTRs showed no significant difference in structural changes compared to random sets.

1.3. The authors should explore whether the DHX36-binding sites exhibit enriched long-range RNA-RNA interactions with distal regions. This analysis could shed light on potential long-range structural rearrangements induced by DHX36 binding.

Thank you for the constructive comment. The Reviewer 2 has a similar suggestion (comment 2.4). As suggested, we have now examined the long-range RNA-RNA interactions using the publicly available PARIS (Psoralen Analysis of RNA Interactions and Structures) data in WT HEK293T cells (GSE74353). As a result, (revised Fig. S2G-I, Text page 9), DHX36-bound

mRNAs were enriched for long-range RNA-RNA interactions across UTRs and CDS at a higher level compared to all mRNAs. We also took a closer look into DHX36 binding sites and found 18.75% DHX36-bound mRNAs harbored DHX36 binding sites overlapped with at least one arm (interactive RNA fragment in PARIS) of the long-range interactions, which is significantly higher than the shuffled sites. These results indicate that DHX36 binding may induce long-range structural rearrangement. In addition, we also discovered 25 DHX36-bound mRNAs with binding sites overlapped with both arms of long-range interactions, of which only 1 harbored 5'UTR-3'UTR interaction, indicating the simultaneous binding of DHX36 on 5'UTR and 3'UTR regions of the same mRNA may not be a consequence of long-range interactions.

1.4. A comprehensive analysis of the sequence and/or structural motifs enriched in the differentially remodeled regions (DRRs) of DHX36 wild-type vs. knockout cells would provide insights into the consequences of DHX36 depletion and potential cellular and physiological effects.

Thanks for your comment. As suggested, we have now performed motif analysis and identified the sequence motifs of the DRRs and found they were most significantly enriched for the G-rich motif as expected (Fig. S3G, Text page 10). Also, the differential structure motifs between WT and DHX36-KO samples were examined by ViennaRNA package, and we found the DRRs in KO samples were differentially enriched for the paired structural motif (revised Fig. S3H and Text page 10), in agreement with the RNA unwinding capacity of DHX36 binding.

1.5. There appears to be confusion regarding the terminology used to describe the loss of RNA structures. For instance, the title of the first subsection suggests a "global loss of RNA structures" induced by DHX36 depletion, but the text indicates that DHX36 depletion leads to structural increase. The authors should clarify and ensure consistency in the usage of terminology throughout the manuscript.

Thank you for the comment. We apologize for the confusion caused by our mistakes in the original manuscript. We have now corrected the title of the first subsection into "global increase of RNA structures". We have also conducted careful editing of the manuscript and ensured the consistent usage of terminology.

1.6. The correlations presented in many figures (e.g., Figure 4F-I, particularly G and H) appear relatively weak (here in Figure 4 indicating subtle effects of DHX36 knockout on most transcripts). It would be beneficial to perform additional analyses to identify subsets of transcripts that are more strongly affected by DHX36 loss.

Thanks for the critical comment. As suggested, we have now performed hierarchical clustering on the DHX36-bound mRNAs based on their abundance changes and the average reactivity changes of the designated regions. As a result (revised Fig. 4J and Text page 15), we identified subsets of mRNAs that are strongly affected by the DHX36-induced structural change. For example, the cluster 2 in the 5'UTR group showed strong negative correlation between structure changes and mRNA abundance changes, whereas the cluster 5 in the 3'UTR group showed positive correlation.

1.7. I suggest investigating the translational changes upon DHX36 loss and exploring the role of RNA structures in this process. Given the involvement of YTHDF1 in translational regulation, it would be interesting to examine its participation in the context of DHX36-mediated effects.

Thank you for the interestingly suggestion. As suggested, we have now analyzed the Ribo-seq data in WT and DHX36-KO HEK293 cells (GSE105175) to investigate the relationships between DHX36-induced translational changes and RNA structural changes; unfortunately, the data quality was too low to support our analysis. We therefore leveraged our previously published polysome profiling data in WT and Dhx36-KO C2C12 cells. By integrating the datasets with the Structure-seq data, we found the mRNAs with 3'UTR structural increase in KO samples showed significantly lower fold changes of translational efficiency (TE) (revised Fig. S7E, Text page 18), indicating DHX36-induced 3'UTR structural decrease may promote mRNA translation. Next, a closer view into the localized structure of the DHX36 binding sites revealed that the mRNAs with 3'UTR DRRs showed larger decrease in TE upon Dhx36 loss (revised Fig. S7F, Text page 18). To investigate the potential role of m6A/YTHDF1 in the DHX36-mediated TE, we found that DHX36-bound mRNAs with down-regulated TEs upon DHX36 loss showed a significantly higher proportion of m6A modification. The above results indicate that DHX36-induced 3'UTR structure loss probably promotes mRNA translation through m6A/YTHDF1, warranting further investigation in the future endeavors.

Reviewer #2:

In this manuscript, Zhang and colleagues propose a new mechanism of regulation by DHX36, and its interplay with m6A. The story is potentially interesting, but it raises a number of questions. I have here summarized the main points:

2.1. High reproducibility was observed: 0.75-0.82. I would like to stop perpetuating the claim made by several groups that correlations below 0.9 are "high correlations" when it comes to chemical probing, because that is definitely not the case. In some cases apparent lower correlations can be due to the fact that certain bases exhibit extremely high reactivities, therefore using a Spearman correlation or discarding outliers above a certain RT-stop intensity, might help revealing the actual correlation. Also, was the correlation calculated by including or excluding the lowly expressed transcripts? Those might negatively affect correlation, but they do not reflect the authentic reproducibility of the experiment.

Thank you for the critical comment. As suggested, we have now performed the analysis by excluding the transcripts with low RT-stop coverages (average coverage per base < 1) and removed the RT-stop outliers (RT stops > 10e5). As a result, we indeed observed the increased correlation coefficients (all correlations are now above 0.9) (Revised Fig. S1A, Text page 6).

2.2. Agreement between rRNA structures and SHAPE data must be evaluated as a AUROC instead.

Thank you for the constructive comment. We have now included the ROC curves and AUC scores to evaluate the agreement between rRNA structures and SHAPE data (revised Fig. S1D-E and Text page 7).

2.3. What statistical test was used in Fig. 1E? The medians do not look different, suggesting that only a tiny fraction of transcripts are actually affected by the loss of DHX36. Was the Gini calculated along the full transcripts? What would happen if breaking down the transcripts into deciles, or simply into 5' UTR, CDS and 3'UTR and performing the same analysis? Same for Fig. 2E.

Thank you for the great comment. In the original Fig. 1E and Fig. 2E, Wilcoxon signed-rank test was used, and the Gini index was calculated along the full transcripts. As suggested, we have now broken down the transcripts into 20 bins (5 for 5'UTR, 10 for CDS, and 5 for 3'UTR) and calculated Gini index for each docile. As a result, the bins at the 3'end showed higher increase in Gini index upon DHX36 loss (revised Fig. 1F and Fig. S3A, Text page 7 and page 9), which is consistent with the binned reactivity changes (Fig. 1G and 2G).

2.4. The binding of DHX36 to both 5UTRs and 3UTRs might be a consequence of the presence of long-range interactions in mRNAs, between these regions. Upon crosslinking the interacting portions would be both captured during CLIP. It is possible to speculate that DHX36 regulates the switch between long-range base-pairing and local G4 formation. In this regard, I would suggest the authors to analyze previously published PARIS in HEK293 cells (Chang's lab) and other RNA duplex capture data to see if these regions are indeed enriched for long-range interactions. The same can also be investigated in terms of base-pairing probabilities using the ViennaRNA package partition function, using either the SHAPE reactivities from the WT or KO cells as constraints, to see if a significant difference in the distribution of base-pair distances exists.

Thank you for the great comment. The reviewer 1 has a similar suggestion in his/her comment 1. 3. As suggested, we have now examined the long-range RNA-RNA interactions using the publicly available PARIS (Psoralen Analysis of RNA Interactions and Structures) data in WT HEK293T cells (GSE74353). As a result, (revised Fig. S2G, Text page 9), DHX36-bound mRNAs were enriched for long-range RNA-RNA interactions across UTRs and CDS at a higher level compared to all mRNAs. We also took a closer look into DHX36 binding sites and found 18.75% DHX36-bound mRNAs harbored DHX36 binding sites overlapped with at least one arm (interactive RNA fragment in PARIS) of the long-range interactions (Fig. S2H), which is significantly higher than the randomly shuffled sites. These results indicate that DHX36 binding may induce long-range structural rearrangement. In addition, we also discovered 25 DHX36-bound mRNAs with binding sites overlapped with both arms of long-range interactions, of which only 1 harbored 5UTR-3UTR interaction (Fig. S2I), indicating the simultaneous binding of DHX36 on 5'UTR and 3'UTR regions of the same mRNA may not be a consequence of long-range interactions.

As suggested, we also compared the base-pairing probabilities (BPPs) between DHX36-bound 5'UTR-3'UTR sites within the same mRNAs and all other bound UTR sites by ViennaRNA

package with SHAPE reactivities from WT and KO cells as constraints. We reason that if long range interactions exist between the DHX36-bound 5'UTR and 3'UTR sites, they should have higher BPPs upon DHX36 loss. However, BPPs for the 5'UTR-3'UTR sites simultaneously bound by DHX36 showed no difference with other UTR sites in both WT and KO (as shown below), supporting that simultaneous DHX36 binding on 5'UTR-3'UTR sites of the same mRNAs may not induce long-range interactions.

Figure. The comparison of BPPs between the DHX36-bound 5'UTR-3'UTR sites on the same mRNAs (dashed lines) and other bound UTR sites (solid lines).

2.5. Is m6A presence necessary for DHX36 binding to RNA? Does DHX36 sense directly the presence of the modification, or does the m6A deposition lead to a structural change that makes DHX36 binding possible? I would encourage the authors to evaluate this (either transcriptome-wide or on a subset of targets), by checking DH36 binding following METTL3/14 knockdown/knockout.

Thanks for the interestingly comment. We would like to point out that our study mainly investigates how DHX36 binding regulates YTHDF1 binding through resolving the structures of m6A sites rather than how m6A impacts DHX36 binding. Nevertheless, it will be interesting to test if m6A presence is necessary for DHX36 binding to RNAs as previous studies showed the colocalization between RNA G4 and m6A (PMID: 32396327 and 30834310). As suggested, we have now performed m6A inhibition in the 293T cells by treating the cells with STM2457 (a METTL3 inhibitor) and followed by RIP-PCR measurement of DHX36 binding as well as Structure-seq analysis. Interestingly, on the 4 identified targets of DHX36, we did not observe consistent changes: DHX36 binding on ZNF768 was reduced but increased on EZR and MEPCE; no significant change was observed on PHF23.

Nevertheless, results from Structure-seq (revised Fig. S7I-L and Text page 19) showed m6A inhibition induced limited structural changes on all 4 targets, indicating that the structural changes are induced by DHX36, rather than m6A.

Reviewer #3:

In this manuscript, Zhang et al. investigate the alteration of the RNA structurome induced by DHX36 depletion. DHX36 is shown to open RNA structures at 3'UTRs, which correlates with decreased RNA abundance. Furthermore, N6-methyladenosine (m6A) modification sites and YTHDF1 binding sites are found to be enriched in DHX36 binding regions. The structural changes in RNA induced by DHX36 may facilitate YTHDF1 binding to m6A sites, subsequently leading to RNA degradation. Overall, this study profiles the changes in the DHX36-induced RNA structurome using NAI detection and demonstrates its impact on RNA abundance through the regulation of YTHDF1 binding. The work is well executed, and the results are presented clearly. However, I do have some questions and suggestions for the authors:

3.1. Fig. 1c demonstrates that NAI probing is unbiased in 5.8s rRNA, as shown by gel analysis. To validate the unbiased nature of RNA structure sequencing data in different treatment groups, the authors should exhibit conserved mRNA elements or rRNA structures in different treatment groups using sequencing data.

Thank you for the critical comment. We have now included the 18S rRNA structures in KO samples along with WT (revised Fig. S1D-E and Text page 7), which validated the unbiased nature of RNA Structure-seq data in different treatment groups.

3.2. In Fig. 1d, the global average SHAPE scores of mRNA are significantly decreased (FC=0.74). In Fig. 2e, it is shown that the fold change (FC) of DHX36-bound mRNA is 0.68. It would be helpful to explain why the RNA structural changes in DHX36-bound mRNA are less pronounced than global changes.

Thank you for the comment. We apologize for the confusion caused by the writing in the original manuscript. In fact, the average fold change of DHX36-bound mRNAs (KO/WT=0.68) is lower than that of all mRNAs (0.74), indicating the structural changes in DHX36-bound

mRNAs are more pronounced than global changes. We have now revised the writing in the revised Text page 7.

3.3. Fig. 2i indicates that the significant structural changes in 3'UTRs are possibly caused by low GC content. However, DHX36 is known to unwind RNA rG4 structures. It would be interesting to explore the reasons behind these extensive structural changes in 3'UTRs, especially in the context of low GC content.

Thanks for the interesting comment. As suggested, we have now discussed the reasons behind the structural changes in 3'UTRs in the context of low GC context (Fig. 2I, revised Text Page 21). Briefly, we think it is possible that low GC high AU content in the 3'UTRs causes low thermostability thus increased susceptibility to the unwinding process. rG4 binding by DHX36 is an ATP-independent process but unwinding is energy consuming which involves a partial unfolding-refolding cycle (PMID: 31015431). Therefore, despite the higher number of G4s in 5'UTRs compared to 3'UTRs and dense DHX36 binding (Fig. S2A-B), the 5'UTR G4s possess higher stability (Fig. S2C) thus may be more resistant to DHX36 unwinding (PMID: 25653156).

3.4. The author's findings suggest more localized structural effects surrounding DHX36 binding sites (Fig. 3c). To strengthen this observation, it would be beneficial to present longer regions that exhibit these effects, rather than just focusing on the +/- 50nt of the crosslink site.

Thank you for the great comment. As suggested, we have now examined regions of various sizes from +/- 50nt to +/- 150nt, increased by +/- 25nt per time, to investigate the reactivity changes. As a result, (revised Fig. S4A, Text page 11), longer regions showed weaker reactivity changes upon DHX36 loss, indicating the structural remodeling effect is confined to small regions and +/- 50nt was thus chosen as the localized sites.

3.5. It would be valuable for the authors to investigate whether DHX36 binds to mRNA JUN. If it does, the binding site should be labeled on Fig. 3h for clarity.

Thank you for the comment. We apologize for the confusion caused by the unclear statement in the original Fig. 3h (Revised Fig. 3H) legend. The folding shown in the figure is the exact DHX36 binding site on mRNA JUN. We have now corrected the labeling.

3.6. Regarding the canonical m6A and YTHDF1 binding motifs, they typically follow the RRACH motif, especially for conserved adenosine. It appears that the DHX36 3'UTR binding sites do not show enrichment for the m6A binding motif (Fig. 3b). the author should do YTHDF1 iCLIP in WT and DHX36 KO groups, and combine RNA structure data/m6A sites, to prove DHX36 regulate YTHDF1 binding by RNA structure switch in m6A-dependent manner.

Thank you for the critical comment. We believe you are referring to our original Fig. 5B not 3B. We think it is possible that the m6A motif may be masked by the dominant G-repeat motifs thus could not be *de novo* discovered in the original analysis. We have now performed m6A/YTHDF1 motif enrichment analysis to replace the *de novo* result (revised Fig. 5B, Text page 16).

In addition, as suggested, we have also performed YTHDF1 eCLIP in WT and DHX36-KO HEK293T cells. As a result, DHX36 loss did not alter the binding preference of YTHDF1 for the GGAUC m6A content (revised Fig. 5I and Fig. S7D, Text page 18) but indeed induced YTHDF1 binding decrease within the DHX36 binding sites (Fig. 5J-K, Text page 18).

3.7. Previous studies have found that m6a can unwind local structures. The author should perform an m6a deletion assay to demonstrate that the structural changes are induced by DHX36, rather than by m6a.

Thank you for the critical comment. The reviewer 2 had a similar suggestion in his/her comment 2. 5. As suggested, we have now treated the cells with a m6A inhibitor, STM2457 and conducted Structure-seq. We found the reduced m6A level did not cause significant structure changes in the vicinity of the m6A sites on the four selected targets, EZR, MEPCE, ZNF768, and PHF34 (revised Fig.S7H-L and Text page 19). The results suggest that the structural changes are induced by DHX36, rather than m6A.

3.8. In Figure 6, the author should conduct reporter gene experiments to validate their findings and molecular mechanisms, rather than solely relying on qPCR to verify these sequencing data.

Thanks for the great comment. As suggested, we have now constructed EGFP reporters by fusing the DHX36 3'UTR binding region sites of each of the four mRNAs downstream of a EGFP reporter. Expectedly, decreased degradation rate and increased stability of the EGFP mRNA were observed in the DHX36-KO vs. WT cells (revised Fig. 6E-F, Text page 19).

Point to point response

Reviewer #1 (Remarks to the Author):

The revision has addressed some of my concerns, but there are still a few outstanding questions.

1.1 I am curious about the impact of the normalization approaches. Simply stating that a script was run is not enough detail. Could you provide more information on how the normalization was effective?

Thank you for the critical comment. We apologize for the unclear description of the normalization process. Briefly, the raw reactivities were calculated using log-normalized RT stops and then normalized by 2-8% methods to generate final reactivities. In the normalization process, the top 10% of raw reactivities were extracted. Within this subset, the top 2% raw reactivities were excluded and the remaining 8% reactivities were averaged to generate a normalization scale. Subsequently, the raw reactivities of both WT and KO samples were divided by the WT normalization scale to generate the normalized reactivities. We have now provided more details in the “Methods” section (Page 33).

To demonstrate the effectiveness of this normalization approach, we compared the SHAPE reactivities of 18S rRNA before and after normalization and found comparable SHAPE reactivities across WT and KO samples upon normalization (Fig. S1B and Page 7 of the Result).

1.2 The authors presented interesting findings in their response. However, I am specifically interested in understanding the structural differences between the DHX36-unbound mRNAs and the non-binding sites/regions within the DHX36-bound mRNAs.

Thank you for the comment. We apologize for our previous misunderstanding. As suggested, we have now performed additional analyses to compare the DHX36-unbound mRNAs and the non-binding sites/regions within the DHX36-bound mRNAs. First, we compared the regional reactivity changes and found that the unbound regions of DHX36-bound mRNAs showed greater structural changes than the same regions of unbound mRNAs (**see below Figure 1A-C**). In addition, we randomly selected 1000 unbound sites from the designated regions of DHX36-bound mRNAs, and 1000 random sites from unbound mRNAs to compare the localized reactivity changes. To ensure reproducibility, we repeated this process 100 times and generated 100 p-values using Wilcoxon rank-sum test (**see below Figure 1D-F**). As a result, all p-values were lower than 0.05, indicating that the random unbound sites of DHX36-bound mRNAs showed higher structural changes than the random sites of unbound mRNAs.

The above results are consistent with our reported finding that DHX36 binding-induced structure remodeling was not confined to localized binding sites (Fig. 2J) but extended across the entire transcript (Fig. 2K-L, Text Page 21).

Figure 1. Structural differences between the DHX36-unbound mRNAs and the non-binding sites/regions within the DHX36-bound mRNAs. (A-C) Comparison of the reactivity changes between the unbound (A) 5'UTRs, (B) CDSs, and (C) 3'UTRs of DHX36-bound mRNAs and the same regions of unbound mRNAs. (D-F) Comparison of the reactivity changes between randomly selected unbound sites from the (D) 5'UTR, (E) CDS and (F) 3'UTR of DHX36-bound mRNAs (blue) and random sites from the same regions of unbound mRNAs (red). The randomization and comparison were performed 100 times. The distribution density of Δ Reactivities and the accumulative proportion (%) of the $\text{log}_2(\text{P-values})$ are shown in the bottom and top panels, respectively. Wilcoxon rank-sum test was used to calculate the significance.

1.6 Is there any additional insight or findings from analyzing the subsets of mRNAs that are most strongly affected by the DHX36-induced structural changes? I'm curious to understand more about the functional roles and regulatory mechanisms associated with these transcripts.

Thank you for the great comment. As suggested, we have now selected the clusters with $|r| > 0.25$ and performed functional enrichment analyses on the mRNAs within these clusters. Gene ontology (GO) analysis indicated that the subsets of mRNAs that are most strongly affected by the DHX36-induced structural changes are enriched in diversified processes, including RNA splicing, RNA processing, rhythmic process and apoptosis et al. We have now included these results on Text Page 15 and a newly generated Fig. S7.

1.7 I appreciate the effort to include the analysis of published polysome profiling data from WT and DHX36-KO C2C12 cells. However, since these are different cell lines, the results are not entirely convincing. I would suggest either removing these results or performing the polysome profiling analysis directly in the HEK293 cells, to provide more meaningful and relevant data.

We agree with your comment. This part of results seems to be distracting and has now been removed from the manuscript.

Please check the items below carefully and add a response in each row of the table to indicate the changes that you have made.

Abstract and editor's summary

Our guidance:

Your response:

Please edit the title so that it is 15 words or fewer and does not include punctuation.	We have removed the slash “/” in the title.
--	--

Author information

Our guidance:

Your response:

We ask that you consult with your coauthors to ensure that all names, affiliations, and titles are represented correctly. Note that if any authors are added or removed after this point then all authors will be requested to provide approval documentation that could potentially delay the production of your paper.	The author information is correct.
Our format allows only two specific author tagging statements, "These authors contributed equally" and "These authors jointly supervised this work", with one of each tag allowed. Please ensure that the statements you provide adhere to these rules.	Our statements adhere to this rule.

Article structure

Our guidance:

Your response:

We can accommodate up to 10 display items (Figures or Tables) in the main article. Each Figure and Table must fit easily within an A4 page (210 x 297 mm). Please ensure that the number and size of your Figures and Tables fulfil these requirements to avoid any delay in the acceptance of your article.	Our display items are less than 10 and they fit easily within an A4 page.
---	--

Please ensure your main manuscript file includes the following sections, in this order:

- Title*
- Author list*
- Affiliations*
- Abstract*
- Introduction*
- Results*
- Discussion (optional)*
- Results and Discussion (optional)*
- Methods*
- Data Availability*
- Code Availability (if relevant)*
- References*
- Acknowledgements*
- Author Contributions Statement*
- Competing Interests Statement*
- Tables*
- Figure Legends/Captions (for main text figures)*

We do not edit Supplementary Information files; they will be uploaded with the published article as they are submitted with the final version of your manuscript. Any tracked changes should be removed from the file and the file should be provided as a PDF file. Supplementary Figures do not need to be provided separately.

Please supply a Source Data file for all data presented in graphs within the Figures.

The sections have been re-ordered.

The source data has been attached.

Please ensure that uncropped scans of all blots and gels in Figures are provided in the Source Data file.

All source data has been provided.

Within the Source Data file, the relevant raw data from each figure or table (in the main manuscript and in the Supplementary Information) should be represented by a single sheet in an Excel document, or a single .txt file or other file type in a zipped folder. An example of the Source Data file is available demonstrating the correct format:

<https://www.nature.com/documents/ncomms-example-source-data.xlsx>

The file should be labelled 'Source Data', with the title and a brief description included in your response here, and should be mentioned in all relevant figure legends using the template text

The source data are now in the correct format.

The source data include:

1. WB gel image: DHX36 protein abundance in WT and DHX36-KO cells.
2. Gel image: sequencing gel for 5.8S rRNA showing the RT stops induced by NAI modification.
3. Ct values of RT-qPCR: mRNA abundance of EZR,

below:

'Source data are provided as a Source Data file.'

A reference to the source data file should be added in the 'Data Availability' section, using the text "Source data are provided with this paper."

MEPCE, PHF23, and ZNF768 in WT and DHX36-KO cells.

4. Ct values of RT-qPCR: mRNA abundance 0h, 4h and 8h after actinomycin D treatment
5. Ct values of RT-qPCR: the abundance of the reporter mRNA fused with DHX36 3'UTR binding site 0h, 2h and 6h after actinomycin D treatment
6. WB gel image: enrichment of the immunoprecipitated YTHDF1 and GAPDH proteins in WT and DHX36-KO cells
7. Ct values of RIP-qPCR: YTHDF1-bound mRNA abundance in WT and DHX36-KO cells.
8. WB gel image: YTHDF1 protein abundance in WT and DHX36-KO cells.

Main text

Our guidance:

Your response:

Please refrain from using words such as new/novel/first, when referring to the scientific findings. Please also remove exaggerated language such as 'extremely'/'outstanding'	Language has been revised.
Please ensure that gene and protein names are formatted according to community standards for the organism in question. Italicize gene symbols and functionally defined locus symbols; do not use italics for proteins.	Gene and protein names are now in correct format.
Please do not use italics, bold font, underlining or speech marks/quotation marks except in headings unless required for technical terms (in both the main text and the display items).	Done.

Please make sure that mathematical terms throughout your manuscript and Supplementary Information (including in figures, figure axes, and legends) conform strictly to the following guidelines. Equations must be supplied in editable format, and not as images. Scalar variables (e.g. x, V, χ) must be typeset in italic, whereas multi-letter variables and functions (e.g. \log) must be formatted in roman. Vectors (such as the wavevector k or the magnetic field vector B) must be typeset in bold without italics.	Done. The mathematical terms are now in the correct format.
Please label equations sequentially as (1), (2), (3), etc.	Equations have been labeled (Page 33).

Figures and Tables

Our guidance:

Your response:

Please see the guidelines linked below for detailed instructions about how your figures should be prepared. Following these instructions will reduce the chances of delays should we need to request replacement artwork from you at a later stage. https://www.nature.com/documents/NRJs-guide-to-preparing-final-artwork.pdf	The instructions have been followed.
Please ensure that data presented in a plot, chart or other visual representation format shows data distribution clearly (e.g. dot plots, box-and-whisker plots, violin plots). When using bar charts, please overlay the corresponding data points (as dot plots) whenever possible and always for $n \leq 10$. All box-plot elements (center line, limits, whiskers, points) should be defined in the legends accompanied by precise n numbers. Please note that data presentation has to be revised to comply with our policy in figure(s) 5c, d; 6c, i.	We added the dataset numbers ($n=100$) of Fig. 5c-d in the legend. As this number is large, it is hard to add data points in the figure. The data points have been added in Fig6c, i.
Wherever statistics have been derived (e.g. error bars, box plots, statistical significance) the legend needs to provide and define the n number (i.e. the sample size used to derive statistics) as a precise value (not a range). Please define how many replicates were performed and whether they are biological (derived from different experimental units or subjects) or technical (multiple contemporary measurements from the same experimental unit or subject). Samples should be unambiguously described, including a clear definition of the unit of study. In studies using model	The sample sizes have been added in the legends. The nature of entity for “n” has been added in the legend of Fig.6c-d, f, i.

organisms, cell lines, primary cell cultures, plants or micro-organisms, the unit of study is the smallest object that could be randomly and independently assigned to an intervention. The groups being compared, including control groups, should be clearly defined. If no control group has been used, the rationale for this should be stated. Please ensure there are enough details about sample collection to distinguish between independent data points and technical replicates- splitting a biological sample into 3 tubes or wells receiving the same treatment does not constitute independent replication.

We strongly discourage deriving statistics from technical replicates or less than 3 biological replicates, unless there is a clear scientific justification for why providing this information is important. Conflating technical and biological variability, e.g., by pooling technically replicates samples across independent experiments is strongly discouraged.

Please note that this information is missing in the legend(s) of figure(s) 1d, e; 2e, h; 4e, i; 5c, d.

Although 'n' is provided, please describe the nature of entity for 'n' in the legend(s) of figure(s) 6c, d, f, i.

All error bars need to be defined in the legends (e.g. SD, SEM) together with a measure of centre (e.g. mean, median). For example, the legends should state something along the lines of "Data are presented as mean values +/- SEM" as appropriate. All box plots need to be defined in the legends in terms of minima, maxima, centre, bounds of box and whiskers and percentile.

Please note that the error bars/error bands need to be defined in the legend(s) of figure(s) 4f-j; 5c, d; 6c, d, f, g, i.

Please note that the box plots need to be defined in terms of minima, maxima, centre, bounds of box and whiskers and percentile in the legend(s) of figure(s) 1d, e (insets); 2e, h (insets); 4e (inset).

We have added descriptions for error bars and boxplots in the legends.

The figure legends must indicate the statistical test used. Where appropriate, please indicate in the figure legends whether the statistical tests were one-sided or two-sided and whether adjustments were made for multiple comparisons. For null hypothesis testing, please indicate the test statistic (e.g. F, t, r) with confidence intervals, effect sizes, degrees of freedom and P values

Two-sided Wilcoxon rank-sum test was used for Fig.4d, which was described together with Fig.4e. The p-values in Fig.5b were generated by HOMER software. The p-values in Fig. 5c-d were generated by Mont

noted. Please provide the test results (e.g. P values) as exact values whenever possible and with confidence intervals noted.

Carlo methods. Therefore, no more description can be provided. The details of p-value calculation and adjustment have been added in the title of Supplementary table 3.

The information on whether the statistical test used was one-sided or two-sided has been added. For example, “The statistical significances in (c-d, f-g, and i) were calculated by two-sided Student’s t-test.” has been added in the legend of fig.6c-d, f-g and i.

Our systems use double-precision floating-point format, which typically provides about 15-17 decimal digits of precision. The p-values smaller than $2.23e-308$ may not be represented accurately. Therefore, $p < 2.23e-308$ was used in Fig 1d.

Please indicate the statistical test used for data analysis and where appropriate, please specify whether it was one-sided or two-sided and whether adjustments were made for multiple comparisons, in the legend(s) of figure(s) 4d; 5b-d; supplementary table(s) 3.

Monte Carlo methods were used in Fig. 5c-d, which compare the true set to 100 random sets. The true set has more m6A/YTHDF1 binding than all 100 random sets, indicating less than 1 event supporting the null hypothesis. Therefore, $p < 1/100 = 0.01$. That is why no exact p-value can be provided for Fig 5c-d.

Please note that the information on whether the statistical test used was one-sided or two-sided, where appropriate, is missing in the legend(s) of figure(s) 1d, e; 2e, f, h, i, m; 4e-j; 5e, g; 6c, d, f, g, i.

Please note that the exact p value should be provided, when possible, in the legend(s) of figure(s) 1d; 5c, d; 6c, d, f, i.

Please state in the legends how many times each experiment was repeated independently with similar results. This is needed for all experiments, but is particularly important wherever results from representative experiments (such as micrographs) are shown. If space in the legends is limiting, this information can be included in

Done.

a section titled “Statistics and Reproducibility” in the methods section. Please revise this information in the legend(s) of figure(s) 1a.	
Please ensure that all blots and gels are accompanied by the locations of molecular weight/size markers. Where it is necessary to crop blots, please ensure that at least one marker position is present. Please supply uncropped and unprocessed scans of the most important blots in the Source Data file or as a supplementary figure in the Supplementary Information. This should be cited once in the Methods section. For an example of presentation of full scan blots, see the Source Data file of https://www.nature.com/articles/s41467-020-16984-1#Sec35 and for more information, please refer to https://www.nature.com/nature-research/editorial-policies/image-integrity. Quantitative comparisons between samples on different gels/blots are discouraged; if this is unavoidable, the figure legend must state that the samples derive from the same experiment and that gels/blots were processed in parallel. Vertically sliced images that juxtapose lanes that were non-adjacent in the gel must have a clear separation or a black line delineating the boundary between the gels. Loading controls (e.g. GAPDH, actin) must be run on the same blot. Sample processing controls run on different gels must be identified as such in the figure legends, and distinctly from loading controls. Figure(s) 1a; 6h, j require(s) molecular weight markers	Done.
Please note that the figure panel 5l is missing although legend for the same has been provided. Please rectify this appropriately.	We have now removed the figure panel 5l.
Shadings or symbols in graphs must be defined in some fashion. We prefer that you use a key within the image; do not include colored symbols in the legend/caption.	We do not have any colored symbols in the legend.
Figure legends/captions should not exceed 350 words. Please shorten by removing detailed methodological information and/or interpretation, or, if appropriate, consider splitting the affected figures in two.	We have now shortened the legends.

All figure legends must include a brief title that summarises the whole figure.	Done.
Every Supplementary Figure must be accompanied by a legend of up to 350 words, referring to all panels, and a brief title that summarises the whole figure.	Done.
Any abbreviations, symbols or colours present in your figures must be defined in the associated legends.	Done.

Data and Code

Our guidance:

Your response:

Nature journals strongly support public availability of data and code. Please deposit the data and code used in your paper into a public data repository, or alternatively, present the data as Supplementary Information. If data can only be shared on request, please explain why in your Data Availability Statement, and also in the correspondence with your editor. Please note that for some data types, deposition in a public repository is mandatory. Any restrictions on sharing of these data types must be clearly indicated in the statement and discussed with the editor. More information on our data deposition policies and available repositories can be found here: https://www.nature.com/nature-research/editorial-policies/reporting-standards#availability-of-data	All sequencing data has been deposited to GEO database and all codes have been uploaded to Github.
All published manuscripts reporting original research in Nature Portfolio journals must include a data availability statement, within the Methods and under the heading 'Data Availability'. The data availability statement must make the conditions of access to the “minimum dataset” that are necessary to interpret, verify and extend the research in the article, transparent to readers. We ask that you don’t use phrases like ‘available on reasonable request’ but instead specify any restrictions to accessing your data as described below. This minimum dataset may be provided through deposition in public community/discipline-specific repositories, custom proprietary repositories or general repositories like Figshare, Zenodo and Dryad. Providing large datasets in supplementary	All data have been uploaded to GEO and the “data accessibility” section has been included in the text.

information is strongly discouraged and the preferred approach is to make data available in repositories. Please see <https://www.springernature.com/gp/authors/research-data-policy/recommended-repositories> for a list of recommended repositories.

If DOIs are provided, we also strongly encourage including these in the Reference list (authors, title, publisher (repository name), identifier, year).

The Data Availability Statement should also reference any source data published alongside the paper.

For clinical datasets or third party data, please ensure that the Data Availability statement adheres to our policy (<https://www.nature.com/nature-research/editorial-policies/reporting-standards#availability-of-data>)

If data are unavailable, please indicate the exact reasons why data cannot be made available in a suitable public repository or upon request, including any conditions related to ethical approval, consent from study subjects, commercial or legal restrictions, etc.

For data that are available under restricted access, the Data Availability statement must specify

- the reasons for access restrictions
- what the restrictions are
- how one can get access to the data
- who to contact to request access
- any restrictions on who the data can be made available to or for which purpose
- the expected timeframe for response to access requests
- for how long the data will be available once access has been granted.

Please use the following template to provide all the information stated above:

The XX data generated in this study have been deposited in the YY database under accession code ZZ [add hyperlink here]. The XX data are available under restricted access for {insert reason}, access can be obtained by {explain how}. The raw XX data are protected and are not available due to data privacy laws. The processed XX data are available at YY. The XX data generated in this study are provided in the Supplementary Information/Source

This template has been used.

Data file. The XX data used in this study are available in the YY database under accession code ZZ [Add hyperlink here].	
Please ensure that all DNA sequencing or RNA-seq data is deposited in an approved, publicly accessible repository listed here (https://www.nature.com/nature-research/editorial-policies/reporting-standards#availability-of-data), and that relevant accession codes are stated in the data availability statement.	Yes, they are all deposited to GEO.
We notice that you have deposited your code in a Github repository, which we fully support. We strongly encourage you in addition to make your code citable by obtaining a DOI for the Github repository in order to provide a permanent reference to the version of the code used in this study and improve reproducibility. This can be done by linking the repository to Zenodo, following the instructions here: https://guides.github.com/activities/citable-code/ Please cite the Github repository in your manuscript text or Code Availability statement and in your reference list: authors, title (this paper), repository name, DOI identifier, year. Alternatively, you can deposit the code in Gigantum or Code Ocean for the same purpose.	We have added a DOI for the Github repository.

Methods

Our guidance:

Your response:

Sufficient details of the experiments must be provided in the Methods section such that they could be reproduced without reference to published papers. Use of the term "as described previously" is not encouraged.	All experimental procedures have been described in sufficient details.
Sequences of oligonucleotides (e.g. primers, RNAi, Crispr), or company names and catalog numbers if reagents are commercial, should be provided in the Methods. If this information is lengthy, it may be provided in Excel format, as separate Supplementary Data, mentioning the file in the Methods.	All information has been provided.
Centrifugation speeds must be reported in x g.	The unit has been changed to x g.
Please rename the Methods section as 'Methods'.	Done.
Please ensure that all accession codes used in this study (new ones AND previously published ones) are listed in the Data Availability statement, together with their corresponding hyperlink.	Hyperlinks have been added.

Use precisely the following format to ensure that the links are permanent:
 CODE [hyperlink] (description if necessary)
 Use full DOI hyperlinks [<http://doi.org/xxxxx>] whenever possible.
 For example:
 5XRN [<http://doi.org/10.2210/pdb5XRN/pdb>]
 1483958 [<https://doi.org/10.5517/ccdc.csd.cc1lt5m6>]
 SRP109982 [<https://www.ncbi.nlm.nih.gov/sra/?term=SRP109982>]
 GSE101099
 [<https://www.ncbi.nlm.nih.gov/geo/query/acc.cgi?acc=GSE101099>]
 NQLW00000000
 [https://www.ncbi.nlm.nih.gov/assembly/GCA_002312845.1/]
 PXD016640
 [<http://proteomecentral.proteomexchange.org/cgi/GetDataset?ID=PX016640>]
 EMD-10857 [<https://www.ebi.ac.uk/pdbe/entry/emdb/EMD-10857>]
 BMRB 28095 [<https://dx.doi.org/10.13018/BMR28095>]

End matter

Our guidance:

Your response:

In the 'Author Contributions' section, kindly differentiate between authors who share the same initials by using either their first or last names.	Done
Nature Portfolio defines Competing Interest (CI) as financial and non-financial interests (including but not limited to funding, employment, stocks, shares, patents, personal or professional relationships with individuals or institutions, and unpaid membership advocacy) that could be perceived to directly undermine the objectivity, integrity, and value of a publication, or could be seen as having an influence on the judgments and actions of authors with regard to objective data presentation, analysis, and interpretation. Please thoroughly review our policy on Competing Interests and include a detailed statement both in your final manuscript file and in our manuscript tracking system. Please ensure the statements are identical in both. Be specific about how each point stated relates to the research and list applicable author initials, and/or patent numbers.	There is no competing interest. The statement had been included.

If there are no competing interests, a negative statement must be included. https://www.nature.com/nature-research/editorial-policies/competing-interests	
Please confirm that all relevant funding awarded to each author is described in the Acknowledgements section. List each grant number, followed by the initials of the author who received it.	All funding information has been included.

Additional Revisions

Our Guidance:

Your Response:

For any Supplementary Figures, please check and confirm that:  * If data is presented as bar charts, individual data points are shown using overlaid dot plots. * The n number (i.e. the sample size used to derive statistics) is provided and defined as a precise value (not a range), using the wording “n=X samples/cells/independent experiments” etc. where applicable. * Any chart axis, error bars, scale bars, molecular weight markers, symbols and colour scales are defined. * Any statistical tests used for data analysis are specified and exact p-values are provided either on the figures themselves, in the legend or in the Source Data file. * Wherever representative data such as blots or micrographs are shown, the legend indicates how many times the experiment was repeated with similar results. * Full uncropped scans of any cropped gel/blot images are provided as an additional Supplementary Figure or in the Source Data file. 	All these issues have been checked and revised if possible.
---	--

Preparing your manuscript files

Our guidance:

Your response:

Unless otherwise stated please limit individual file sizes to approximately 30MB. We strongly encourage the use of repositories for large datasets or source data due to size considerations.	All files meet the requirement.
--	--

Please supply a brief (maximum 250 characters, including spaces) summary of the main findings of the paper to be used on our website and in our e-alerts. The summary should be written in the third person in language suitable for a broad audience. The summary may be edited by the editors prior to publication. Please provide this summary in your cover letter.	Done.
Large datasets exceeding an A4 page size should be supplied as Supplementary Data files to allow reuse, not Supplementary Tables. * Supplementary Table X should be Supplementary Data X.	Done.
Please note that all Supplementary Information must be provided as a single separate PDF file, not within the manuscript file. All Supplementary Information items (e.g. Supplementary Figures, Supplementary Tables, Supplementary Methods, Supplementary Notes, Supplementary Discussion, Supplementary References) must be included in one PDF document. Please refer to our formatting guide when preparing your supplementary information file: https://www.nature.com/documents/ncomms-formatting-instructions.pdf All Supplementary Information files (e.g. Supplementary Data, Supplementary Software, etc.) must be cited in the main text. Every Supplementary Figure must be accompanied by a legend of up to 350 words, referring to all panels, and a brief title that summarises the whole figure. Only Supplementary Movie, Audio, Data and Software files should be submitted separately from the Supplementary Information.	The supplementary information is now provided as a separated file.
Please supply legends for each Supplementary Movie/Audio/Data file in your response here (not in the Supplementary Information file). Please label each files as Supplementary Movie/Audio/Data 1, etc.	Data 1. Structure-seq data analyses in HEK293T cells. Data 2. DHX36 binding profiles in HEK293T and C2C12 cells. Data 3. DHX36-induced DRRs within DHX36-bound mRNAs.

The significance was calculated by dStruct. The false positive discovery rate (FDR) method was used to correct for multiple comparisons.

Data 4. Variance inflation factors (VIF) of the explanatory variables in the linear regression models.

Data 5. Structure-seq data analyses in C2C12 cells.

Data 6. Identification of DHX36 post-transcriptional regulatory targets.

Data 7. Identification of mRNA clusters with strong correlation between mRNA abundance and structural changes within the designated regions.

Data 8. m⁶A and YTHDF1 binding sites within the DHX36 binding sites located in 3UTRs.

	Data 9. Sequences of DNA, RNA oligos and peptide used in this study.
All Supplementary Information items (e.g. Supplementary Figures, Supplementary Tables, Supplementary Methods, Supplementary Notes, Supplementary Discussion, Supplementary References) must be included in one PDF document. Only Supplementary Movie, Audio, Data and Software files should be submitted separately from the Supplementary Information.	Done.
The use or adaptation of previously published images is strongly discouraged. If this is unavoidable, please request the necessary rights documentation to re-use such material from the relevant copyright holders and return this to us when you submit your revised manuscript. Please check whether your manuscript or Supplementary Information contain third-party images, such as figures from the literature, stock photos, clip art or commercial satellite and map data. If you have used content from BioRender please provide us with a copy of your license. Please also follow the link below to request the appropriate form from BioRender for publication of their figures. https://help.biorender.com/en/articles/8601313-creative-commons-licensing-for-biorender-figures-premium-only Please note a CC-BY permission form must be obtained from BioRender if your work is due to be published under a Creative Commons (CC) license and you must adhere to the attribution guidelines as set out by the license. For more information on what constitutes ownership by a third party, please contact our Editorial Assistant at naturecommunications@nature.com Please check in particular: Please note that suspected third party content is present in figures 1-(b), 2-(a), 4-(a), 5-(a), 7.	We have uploaded the publication licenses from biorender as supplementary files (Fig 2a, 4a, 5a and 7).

Forms to complete

Our guidance:

Your response:

Editorial Policy Checklist Please update and upload a final version of the Editorial Policy Checklist with your revised manuscript files. A blank Editorial Policy Checklist can be found via the link below. Note that this form is a dynamic 'smart pdf' and must be downloaded and completed in Adobe Reader. Please update your current checklist or download from: https://www.nature.com/documents/nr-editorial-policy-checklist.zip	The checklist has been updated and uploaded.
Reporting Summary Please revise the Reporting Summary according to the requests below. After making the requested changes, please be sure to include the final version of your Reporting Summary in your submission as a supplementary information file. Please note that this form is a dynamic 'smart pdf' and must therefore be downloaded and completed in Adobe Reader, instead of opening it in a web browser. Please update your current checklist or download from: https://www.nature.com/documents/nr-reporting-summary.pdf	The reporting summary has been updated and uploaded.

Reporting Summary

Our Guidance:

Your Response:

Software	
Please ensure all the data collection/data analysis software/tools/algorithms/packages used in the study are clearly mentioned in the manuscript and are also listed in the reporting summary (with version numbers).	All software/tools have been listed.
Code	
Please provide the GitHub web-link for the custom code developed in the study, in the reporting summary as well.	The link to our GitHub repository has been provided.
Data Availability	

Please provide a complete data availability statement in the manuscript and in the reporting summary.	Done.
Please note that the following accession numbers are currently private: GSE237160, GSE237161, GSE264498, and GSE264642. Please ensure that this data is publicly released by the time you resubmit your final manuscript; we will not be able to accept your manuscript if the data is not publicly available.	We have released all data.
Field Specific Reporting	
Life Sciences Study Design	
Please describe the general measures taken to verify the reproducibility of the experimental findings. If all attempts at replication were successful, confirm this OR if there are any findings that were not replicated or cannot be reproduced, note this and describe why.	All attempts at replication were successful.
For all experiments, please describe whether the investigators were blinded to group allocation during data collection and/or analysis. If blinding was not possible, describe why OR explain why blinding was not relevant to your study.	The investigators were blinded to group allocation during data collection.
Please describe how sample sizes were determined (for all experiments performed in this study) and provide a rationale for how they were chosen, detailing any statistical methods used to predetermine sample size and/or including relevant citations if available. If no sample size calculation was performed, describe how sample sizes were chosen and provide a rationale for why these sample sizes are sufficient.	All sequencing assays have at least two biological replicates and all experimental validation have at least 3 biological replicates.
Please clearly describe how samples were allocated into experimental groups. If allocation was not random, describe how covariates were controlled OR if this is not relevant to your study, explain why.	Random allocation.
Antibodies	
Please specify the source and amount/dilution of all antibodies used. For commercially sourced antibodies please state the supplier name, catalog number, clone name, lot number as applicable in both the manuscript and relevant section of the Reporting Summary.	The source and amount of all antibodies have been listed in both the manuscript and the reporting summary.

Please provide information on antibody validation in the reporting summary for all the primary antibodies. Please refer to the specific section of the manuscript, previous literature or manufacturer websites as applicable.	The information on antibody validation is in the reporting summary.
Eukaryotic cell lines	
Please describe the authentication procedures for each cell line used in the reporting summary and manuscript. Please specify the techniques used (STR profiling, Karyotyping, DNA barcoding, PCR assays with species-specific primers, etc). Alternatively, declare which of the cell lines used were not authenticated.	Cell lines were procured from commercial source, which has been described in both the reporting summary and the manuscript.

You will need to upload:

Editorial Policy Checklist	
Completed Third Party Rights Table (if relevant)	
A completed copy of this checklist	
The main manuscript file in either Microsoft Word or LaTeX format	
Separate Figure files (each of the main figures)	
A Source Data file	
Inventory of Supporting Information	
A Supplementary Information file	
Supplementary Data files